# Actin foci facilitate activation of the phospholipase C-γ in primary T lymphocytes via the WASP pathway

Sudha Kumari[1,2]*, David Depoil[1,3], Roberta Martinelli[4], Edward Judokusumo[5], Guillaume Carmona[2], Frank B Gertler[2], Lance C Kam[5], Christopher V Carman[4], Janis K Burkhardt[6], Darrell J Irvine[2], Michael L Dustin[1,3]*

[1]Skirball Institute of Biomolecular Medicine, New York University School of Medicine, New York, United States; [2]David H. Koch Institute for Integrative Cancer research, Massachusetts Institute of Technology, Cambridge, United States; [3]Nuffield Department of Orthopedics, Rheumatology and Musculosceletal Sciences, University of Oxford, Headington, United Kingdom; [4]Beth Israel Deaconess Medical Center, Harvard Medical School, Boston, United States; [5]Department of Biological Engineering, Columbia University, New York, United States; [6]Department of Pathology and Laboratory Medicine, Perelman School of Medicine, The Children's Hospital of Philadelphia, University of Pennsylvania, Philadelphia, United States

**Abstract** Wiscott Aldrich Syndrome protein (WASP) deficiency results in defects in calcium ion signaling, cytoskeletal regulation, gene transcription and overall T cell activation. The activation of WASP constitutes a key pathway for actin filament nucleation. Yet, when WASP function is eliminated there is negligible effect on actin polymerization at the immunological synapse, leading to gaps in our understanding of the events connecting WASP and calcium ion signaling. Here, we identify a fraction of total synaptic F-actin selectively generated by WASP in the form of distinct F-actin 'foci'. These foci are polymerized de novo as a result of the T cell receptor (TCR) proximal tyrosine kinase cascade, and facilitate distal signaling events including PLCγ1 activation and subsequent cytoplasmic calcium ion elevation. We conclude that WASP generates a dynamic F-actin architecture in the context of the immunological synapse, which then amplifies the downstream signals required for an optimal immune response.

*For correspondence: michael. dustin@kennedy.ox.ac.uk (MLD); kumars04@mit.edu (SK)

**Competing interests:** The authors declare that no competing interests exist.

## Introduction

Upon encountering antigen-presenting cells (APCs) displaying T cell receptor (TCR) ligands and adhesion molecules, T cells undergo a series of actin dependent shape changes and signaling steps. These ligands include MHC complexed with antigentic peptide (MHCp) for mouse monoclonal T cells, agonist anti-CD3 antibody for polyclonal mouse (2C11) or human (OKT3) T cells. All of these ligands when presented with ICAM1 in supported lipid bilayers (SLB) trigger a series of changes in T cell synaptic cytoskeleton and morphology (*Campi et al., 2005*; *Kaizuka et al., 2007*; *Burkhardt et al., 2008*; *Ilani et al., 2009*; *Beemiller et al., 2012*; *Kumari et al., 2012*). First, TCR signaling is initiated at F-actin-based protrusions of polarized T cells (*Valitutti et al., 1995*; *Negulescu et al., 1996*). Next, these early activation events lead to F-actin dependent formation of a broad immunological synapse (IS) (*Wülfing and Davis, 1998*; *Grakoui et al., 1999*) containing T cell antigen receptor (TCR) microclusters (MCs) (*Bunnell et al., 2002*; *Campi et al., 2005*). Synapse spreading is followed by myosin II dependent contraction to form supramolecular activation clusters (SMACs) (*Monks et al., 1998*;

**eLife digest** The immune system is made up of several types of cells that protect the body against infection and disease. Immune cells such as T cells survey the body and when receptors on their surface encounter infected cells, the receptors activate the T cell by triggering a signaling pathway.

The early stages of T cell receptor signaling lead to the formation of a cell–cell contact zone called the immunological synapse. Filaments of a protein called F-actin—which are continuously assembled and taken apart—make versatile networks and help the immunological synapse to form. F-actin filaments have crucial roles in the later stages of T cell receptor signaling as well, but how they contribute to this is not clear. Whether it is the same F-actin network that participates both in synapse formation and the late stages of T cell receptor signaling, and if so, then by what mechanism, remains unknown.

The answers came from examining the function of a protein named Wiscott-Aldrich Syndrome Protein (WASP), which forms an F-actin network at the synapse. Loss of WASP is known to result in the X-linked Wiscott-Aldrich Syndrome immunodeficiency and bleeding disorder in humans. Although T cells missing WASP can construct immunological synapses, and these synapses do have normal levels of F-actin and early T cell receptor signaling, they still fail to respond to infected cells properly.

Kumari et al. analyzed the detailed structure and dynamics of actin filament networks at immunological synapses of normal and WASP-deficient T cells. Normally, cells had visible foci of newly polymerized F-actin directly above T cell receptor clusters in the immunological synapses, but these foci were not seen in the cells lacking WASP. Kumari et al. found that the F-actin foci facilitate the later stages of the signaling that activates the T cells; this signaling was lacking in WASP-deficient cells.

Altogether, Kumari et al. show that WASP-generated F-actin foci at immunological synapses bridge the early and later stages of T cell receptor signaling, effectively generating an optimal immune response against infected cells. Further work will now be needed to understand whether there are other F-actin substructures that play specialized roles in T cell signaling, and if foci play a related role in other cell types known to be affected in Wiscott-Aldrich Syndrome immunodeficiency.

Griffiths et al., 2001; Ilani et al., 2009). TCR MCs form in the outer lamellipodia-like 'distal' (d)SMAC, after which signaling MCs move centripetally through the lamella like 'peripheral' (p)SMAC, driven by centripetal flow of F-actin (Ponti et al., 2004; Varma et al., 2006; DeMond et al., 2008; Vardhana et al., 2010; Kumari et al., 2012; Yi et al., 2012) to eventually reach the 'central' (c)SMAC. The cSMAC is an F-actin depleted zone (Stinchcombe et al., 2006), in which TCR is destined for down-regulation via extracellular vesicle formation (Vardhana et al., 2010; Choudhuri et al., 2014).

Actin polymerization and remodeling continues throughout the lifetime of the immunological synapse, and studies using a variety of T cell activation systems have identified a critical requirement for F-actin and its molecular effectors in optimal TCR signaling and T cell function (Valitutti et al., 1995; Holsinger et al., 1998; Snapper et al., 1998; Campi et al., 2005; Burkhardt et al., 2008). Pharmacological disruption of global F-actin or genetic manipulations that reduce synaptic F-actin polymerization result in defects in early signaling events such as cell adhesion, TCR MC formation, early TCR signaling, TCR MC transport, as well as the TCR-distal events including intracellular calcium rise, and store operated calcium ion entry (DeBell et al., 1992; Valitutti et al., 1995; Campi et al., 2005; Nolz et al., 2006; Varma et al., 2006). Since global actin perturbation impairs all the above steps, it has been a challenge to dissect the mechanistic role of F-actin in TCR signaling and to identify whether there exist functionally distinct F-actin networks within SMACs that may play distinct roles at various stages of the pathway.

One way to investigate functional diversity within the subsynaptic F-actin network, during TCR signaling, is to utilize actin perturbations that dissociate early TCR signaling from late TCR signaling. One such context is provided by the loss of an actin effector protein—Wiscott Aldrich Syndrome Protein (WASP). F-actin polymerization relies on regulatory factors such as nucleation promoting factors (NPFs) and downstream nucleation factors such as Arp2/3 complex (Blanchoin et al., 2000).

WASP is a hematopoietic-cell-specific NPF activated downstream of TCR, originally identified as a target of loss-of-function mutations in the X-linked immunodeficiency Wiskott-Aldrich syndrome (WAS) (*Derry et al., 1994*; *Machesky and Insall, 1998*; *Takenawa and Miki, 2001*). WAS T cells exhibit a variety of T cell activation defects such as aberrant synapse morphology, impaired proliferation in response to TCR-activation stimuli, and impaired calcium ion signaling (*Molina et al., 1993*; *Cianferoni et al., 2005*; *Calvez et al., 2011*). Similar to WAS patients, T cells from *Was−/−* mice also display profound defects in antigen receptor-induced proliferation, IS stability, nuclear NFAT translocation and IL-2 production (*Snapper et al., 1998*; *Zhang et al., 1999*, *2002*; *Cannon and Burkhardt, 2004*). T cells from *Was−/−* mice (*Zhang et al., 1999*; *Krawczyk et al., 2002*; *Cannon and Burkhardt, 2004*; *Sims et al., 2007*) and human WAS T cells (*Molina et al., 1993*; *Dupre et al., 2002*; *Calvez et al., 2011*) have apparently normal total F-actin levels as well as SMAC organization within the immunological synapse, while initial TCR–associated kinase signaling in response to MHC-peptide complexes in the context of adhesion ligands is also intact (*Rengan et al., 2000*; *Sato et al., 2001*; *Krawczyk et al., 2002*; *Cannon and Burkhardt, 2004*; *Sims et al., 2007*). Despite many years of study, the F-actin network to which WASP contributes, and the specific TCR-signaling steps in which it participates to regulate calcium signaling, remain unknown.

How might WASP regulate T cell calcium ion responses without affecting total synaptic F-actin? As an NPF, WASP binds to Arp2/3 and G-actin, increasing the ability of Arp2/3 to nucleate actin branches from existing filaments. Moreover, WASP binds hematopoietic lineage cell-specific protein 1 (HS1) through its SH3 domain (*Dehring et al., 2011*). HS1 is also activated in response to TCR stimulation (*Taniuchi et al., 1995*; *Gomez et al., 2006*) and can weakly activate Arp2/3 complex, as well as stabilize branched F-actin filaments (*Weaver et al., 2001*). HS1 deficient T cells show defects similar to WASP−/− T cells in TCR activation dependent calcium elevation, proliferation, IL-2 secretion and NFAT activation (*Taniuchi et al., 1995*; *Hutchcroft et al., 1998*; *Gomez et al., 2006*). It is therefore possible that a previously uncharacterized subclass of the synaptic F-actin network at the TCR MC that represent a small fraction of total synaptic F-actin, is generated by WASP and stabilized by HS1, supports calcium signaling. Alternatively, it has also been proposed that WASP is a modular scaffolding protein capable of interacting with other proteins of the TCR signalosome, independent of its role as an NPF (*Huang et al., 2005*). Although these two hypotheses are not mutually exclusive, an F-actin dependent role could be addressed by identifying the F-actin network in the immunological synapse to which WASP contributes, and independently targeting this network to investigate the role of the WASP-generated F-actin subpopulation in calcium signaling at the synapse. Thus, WASP can be utilized as a tool to probe for functionally distinct organizational categories of F-actin within the synapse.

The signaling cascade leading up to calcium ion elevation in response to TCR engagement has been studied in much detail (*Braiman et al., 2006*; *Mingueneau et al., 2009*; *Sherman et al., 2011*). TCR ligation triggers a molecular program that results in activation of phospholipase C-$\gamma$1 (PLC$\gamma$1), through phosphorylation on Y-783 by Itk (*Park et al., 1991*). Once it has been activated, phospho-PLC$\gamma$1 catalyzes the conversion of phosphatidylinositol-4,5 bisphosphate (PIP$_2$) to inositol tri-sphosphate (IP$_3$) and diacylglycerol. IP$_3$ then acts as a second messenger and facilitates release of calcium ions from intracellular stores. Following TCR activation, PLC$\gamma$1 recruitment at the synapse is primarily mediated via binding to linker of activated T cells (LAT) (*Braiman et al., 2006*). Additionally, recent studies using Jurkat T cells and thymocytes have reported a role for the cortical cytoskeleton in both promoting and inhibiting PLC$\gamma$1 activation (*Babich et al., 2012*; *Tan et al., 2014*). Although PLC$\gamma$1 binds F-actin in biochemical assays, and loss of F-actin dynamics led to reduced PLC$\gamma$1 phosphorylation in Jurkat T cells (*DeBell et al., 1992*; *Carrizosa et al., 2009*; *Patsoukis et al., 2009*; *Babich et al., 2012*), the dependence of PLC$\gamma$1 activation on WASP activity has not been tested in primary T cells. We hypothesize that WASP and HS1 generate an F-actin network that maintains phosphorylation of PLC$\gamma$1 at the synapse, accounting for their role in calcium ion elevation (*Carrizosa et al., 2009*).

In this study, we tested these hypotheses by characterizing the F-actin microarchitecture at the immunological synapse that is selectively regulated by WASP, and evaluating its role in early signaling, HS1 and PLC$\gamma$1 dynamics, and calcium signaling at the immunological synapse. The results presented here identify and functionally characterize a WASP-dependent actin network at the immunological synapse that regulates phospho-PLC$\gamma$1 levels at TCR MC and calcium ion elevation in T cells. This

network is visualized as F-actin foci that result from new F-actin actin polymerization at TCR MC. Disruption of these actin foci does not impair initial synapse formation or early TCR signaling. Importantly, disruption of the F-actin foci using a selective Arp2/3 complex inhibitor also spares early TCR signaling, but results in the same impairment in HS-1 and PLC-γ1 activation that has been observed in WASP deficient T cells. Defining a new F-actin network in the immunological synapse, and its molecular regulation, furthers our mechanistic understanding of the cytoskeletal regulation of T cell activation and its dysfunction in WAS pathology.

## Results

### WASP generates F-actin foci in the immunological synapse

Utilizing a planar surface functionalized with anti-mouse CD3 antibody and ICAM1 as an antigen-presenting surface, we first compared the F-actin cytoskeleton in polyclonal primary CD4 T cells from either wild type (WT) or WASP deficient C57BL/6J mice at the early spreading stage of immunological synapse formation. To resolve dynamic F-actin networks, we employed a labeling strategy designed to highlight growing filament ends (*Furman et al., 2007*). A 1 min pulse of Rhodamine-ATP-G-actin in live-permeabilized cells identified freshly incorporated subunits (Fresh F-actin), while subsequent fixation and Alexa488-phalloidin staining identified total F-actin. T cells synapses were visualized using total internal reflection fluorescence microscopy (TIRF) to selectively illuminate structures within 200 nm of the interface. WT T cells exhibited discrete F-actin rich features (*Figure 1A*, arrowheads). We refer to these structures as F-actin 'foci'. To selectively extract and quantify F-actin foci from the lamellar actin background, we utilized a spatial frequency-filter based local background correction method (*Figure 1A* graph, *Figure1—figure supplement 1A,2*, 'Materials and methods'). This foci extraction method is capable of reliably identifying foci on a variable background of lamellar F-actin (*Figure 1—figure supplement 1B*). Assessment of the amount of newly polymerized actin in foci vs surrounding lamellar areas using the above-mentioned labeling and quantification method revealed that the F-actin foci incorporate fresh actin subunits (*Figure 1B*). This indicates that the foci are dynamically polymerizing structures, rather than sites where F-actin accumulates due to rearrangement of preexisting filaments. Furthermore, the polymerization rate of individual filaments (Fresh/Total ratio) is similar between foci and surrounding areas in WT T cells (*Figure 1B*), indicating that the foci result from preferential local nucleation, rather than enhanced polymerization rate of existing filaments. When compared at the whole synapse scale, overall synaptic F-actin levels, as well as the rate of actin polymerization were similar in WT and WASP−/− T cells, as expected (*Cannon and Burkhardt, 2004*). However, there was a significant reduction in F-actin foci in WASP−/− cell synapse (*Figure 1C*), suggesting that dynamic F-actin foci are formed in a WASP dependent manner. The loss of foci in WASP−/− T cells is not due to non-specific developmental defects, since acute siRNA mediated knockdown of WASP in activated mouse CD4 T cells resulted in a similar phenotype to the WASP−/− T cells. No reduction in total F-actin and significant reductions in foci were observed in WASP siRNA transfected cells imaged on SLB with anti-CD3 and ICAM1 (*Figure 1—figure supplement 3*). This lack of effect on total F-actin in WASP deficient or depleted cells may be attributed to the fact that these foci contribute to <10% of the total synaptic F-actin intensity (*Figure 1—figure supplement 1B* top graph), or perhaps to a compensatory increase in the lamellar and lamellipodial actin when more ATP-G-actin is available for polymerization regulated by other NPFs such as WAVE2. These results indicate that WASP dependent F-actin foci form in the T cell synapse.

We next assessed the role of other WASP-family proteins—NWASP and WAVE2—in generating F-actin foci in WT cells. NWASP (WASL) is a close homolog of WASP and is capable of compensating for WASP's function in T cell development (*Cotta-de-Almeida et al., 2007*). However, primary CD4 T cells from *Wasl−/−* mice (*Figure 1D*) formed immunological synapses with F-actin foci similar to T cells from WT mice. WAVE2 is required for immunological synapse formation itself (*Nolz et al., 2006*), thus we could not examine synaptic foci in WAVE2 deficient T cells. Instead, we determined the localization of NWASP and WAVE2 in the immunological synapse in order to gain further insight into their possible role in foci formation. Activation of T cells with anti-CD3 and ICAM1 recruited NWASP and WAVE2 to the IS (*Figure 1—figure supplement 4A*) (*Nolz et al., 2006*), but the recruited proteins failed to co-localize with F-actin foci (*Figure 1—figure supplement 4B*, NWASP coloc. = 9.48% ± 0.76 n = 78; WAVE2 coloc. = 14.36% ± 1.25, n = 54). We were unable to identify

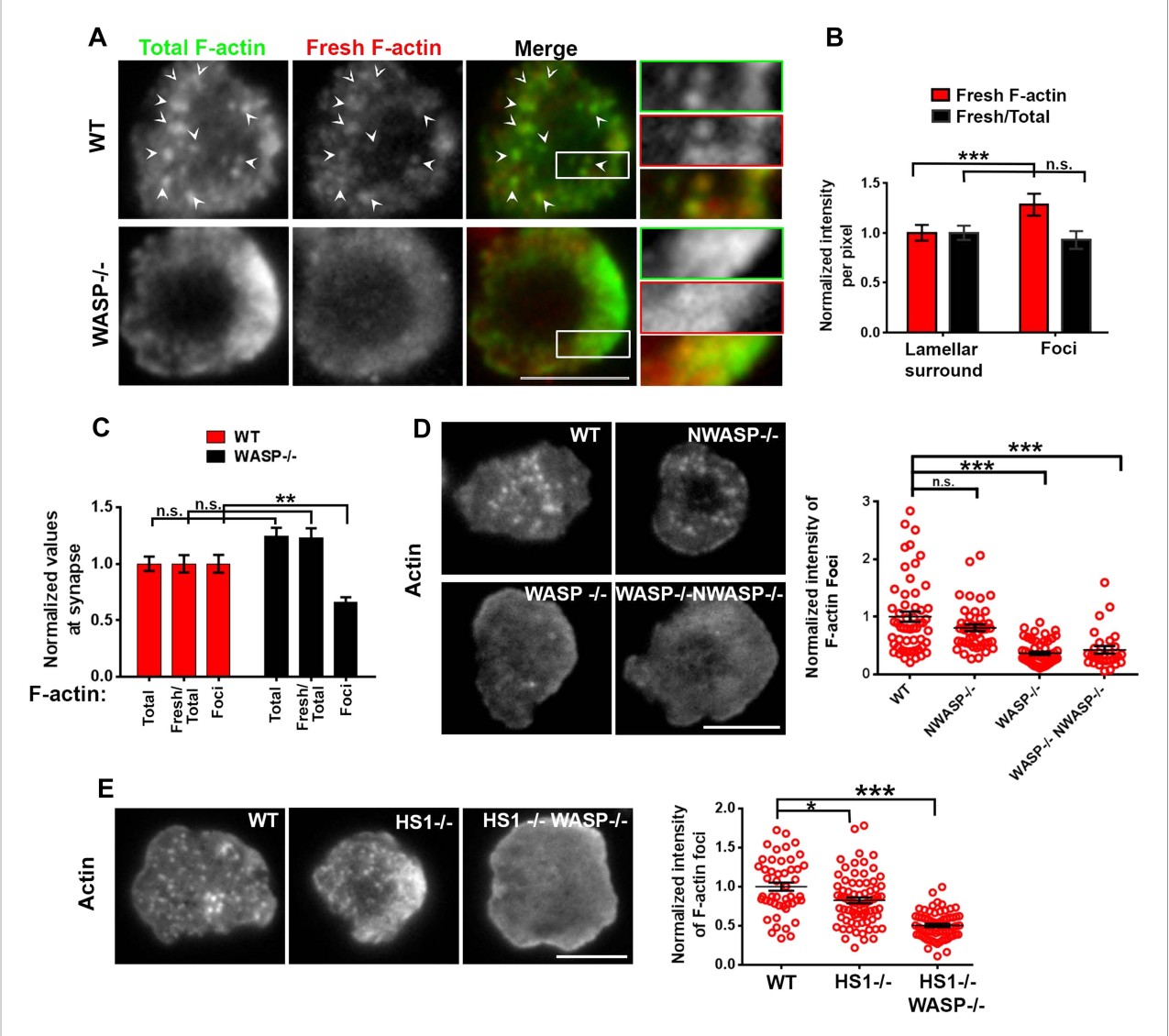

**Figure 1**. WASP dependent dynamic F-actin foci at the T cell synapse. (**A**) Barbed-end decoration of freshly polymerized actin filaments reveals the dynamic behavior in WT (top panel) and WASP−/− (bottom panel) T cell synapses. CD4 T cells isolated from WT and *Was−/−* C57BL/6J mice were processed for barbed end labeling (fresh actin) to identify the actin incorporation sites within 1 min of polymerization, as well as total F-actin labeling, as described in 'Materials and methods'. Arrowheads indicate the sites of F-actin foci in both 'Total' as well as 'Fresh' F-actin images. (**B**) The graph shows average incorporation of Rhodamine actin per pixel within the foci or the surrounding lamellar pixels (Fresh F-actin); and a ratio of Rhodamine and Alexa488 actin intensities in foci or surrounding pixels, in the WT cells. The foci areas in the total F-actin image were identified and outlined by intensity rank-based filtering as described in 'Materials and methods'. The synaptic area outside the foci was defined as lamellar surround. These outlined areas were then analyzed in both 'fresh F-actin', and 'total F-actin' raw images and per pixel intensity is plotted in the graph. Note that while the rate of polymerization (Fresh/Total) ratio is not altered, there is higher incorporation of actin at foci sites (Fresh). n = 20 cells, p1 < 0.0001, p2 = 0.181 (Wilcoxon nonparametric pairwise comparison). (**C**) The graph shows total F-actin intensity, fresh F-actin intensity, or the ratio of the two at the synapse, each point represents value obtained from single cell, n in WT = 33, n in WASP−/− = 35; *p1 = 0.06, p2 = 0.059, p3 = 0.0008*. Scale bars, 5 µm. (**D**) WASP is critical for F-actin foci generation. Freshly purified CD4 T cells from WT 129 (top left), *Was−/−* 129 (bottom left), *Wasl−/−* (top right), or *Was/− Wasl−/−* 129 (bottom right) mice were activated on SLB containing ICAM1 and anti-CD3 antibody, fixed and stained with Alexa488-phalloidin. Note that only the lack of WASP, and not N-WASP, causes loss of actin foci. n1 = 54, n2 = 45, n3 = 53, n4 = 28. *p1 = 0.30, p2 < 0.0001, p3 < 0.0001*. (**E**) F-actin foci in CD4 T cells freshly purified from wild type (WT) mouse (left) or from *Hcls1−/−* mouse (center) or *Hcls1−/− Was−/−* (right) C57BL/6J mice were incubated with bilayer containing anti-CD3 antibody and ICAM1 for 2 min. Cells were then processed for F-actin staining (Alexa488-phalloidin) and visualized. Note that, while there is a minor alteration in F-actin foci in HS1−/−

*Figure 1. continued on next page*

*Figure 1. Continued*

T cells, there is a gross deficit in HS1−/− WASP−/− double knockout T cells. The graph shows the quantification of total intensity of F-actin foci at the synapse in individual T cells derived from the indicated backgrounds. n1 = 48, n2 = 70, n3 = 64, p1 = 0.031, p2 = 0.0001.

The following figure supplements are available for figure 1:

**Figure supplement 1**. Method of analyzing F-actin foci from the raw F-actin TIRF image.

**Figure supplement 2**. Test of rank-filter based processing method for foci detection.

**Figure supplement 3**. WASP silencing causes a reduction in F-actin foci.

**Figure supplement 4**. WASP family members NWASP and WAVE2 are not associated with F-actin foci.

**Figure supplement 5**. WASP dependent HS1 recruitment and F-actin foci.

suitable reagents for staining of endogenous WASP in the immunological synapse. These results demonstrate that NWASP does not contribute to F-actin foci and suggest that WAVE2 is unlikely to contribute directly to formation of F-actin foci.

In addition to WASP, deficiency of HS1, a cortactin-related NPF, also leads to reduced calcium ion signaling in T cells (*Gomez et al., 2006*). Therefore, we examined the role of HS1 in generation of F-actin foci using T cells from WT and HS1−/− mice. F-actin foci formation was only marginally impaired in HS1−/− T cells, this reduction was modest compared to T cells from mice lacking both HS1 and WASP (*Figure 1E*). A similar marginal reduction in F-actin foci intensity was obtained by siRNA-mediated knockdown of HS1 in activated mouse CD4 T cells (*Figure 1—figure supplement 5A*). HS1 is phosphorylated on Y397 (phospho-HS1) during T cell activation (*Hutchcroft et al., 1998*; *Gomez et al., 2006*; *Carrizosa et al., 2009*). We next determined if this phosphorylation is WASP dependent. WT mouse CD4 T cells activated with anti-CD3 and ICAM1 displayed foci of phospho-HS1 in the synapse (*Figure 1—figure supplement 5B*). These phospho-HS1 foci were significantly reduced in WASP−/− or WASP−/− NWASP−/− CD4 T cells, but not in CD4 T cells from NWASP deficient mice (*Figure 1—figure supplement 5B*). Thus, HS1 is recruited to the T cell synapse in a WASP dependent manner.

## F-actin foci localize at TCR MC sites and require TCR-proximal signaling

To understand the mechanism underlying formation of the F-actin foci, we determined if they were associated with TCR MCs. In murine polyclonal CD4 T cells activated on anti-CD3 and ICAM1 containing bilayers, as in *Figure 1*, TCR MC localized with F-actin foci (*Figure 2A*, upper panels, arrows). Similar co-localization of F-actin foci with TCR MC was observed with monoclonal AND TCR transgenic T cells stimulated with MHCp and ICAM1 (*Figure 2A*, lower panels). The tracking of TCR with H57 Fab allowed accurate quantification of co-localization between TCR and F-actin. After 1 min of cell attachment, 40 ± 1.7% of TCR MCs co-localized with F-actin foci in these cells (*Figure 2—figure supplement 1A*, 'Materials and methods', *Figure 2B*), significantly above chance level co-localization (*Figure 2—figure supplement 1B*), whereas only 15 ± 0.07% of total ICAM1 exhibited association with actin foci (*Figure 2B*), which was not different from chance. When examined in Human primary CD4 T cells as well, F-actin foci formed in WASP-dependent manner (*Figure 2—figure supplement 2A,B*), and TCR MCs, but not ICAM1 MCs, were associated with F-actin foci (TCR co-localization with F-actin foci was 34.5% ± 2.0, n = 76, co-localization of ICAM1 with F-actin foci was 13.1% ± 0.8, n = 66) (*Figure 2C,D*). Furthermore, TCR-activation led to HS1 phosphorylation at the T cell synapse, as shown previously (*Figure 2—figure supplement 3A*) (*Hutchcroft et al., 1998*; *Gomez et al., 2006*), and the phospho-HS1 co-localized with F-actin foci in both TCR-activation stimuli (*Figure 2—figure supplement 3B*). This indicated that TCR-MCs associated foci exist in a variety of primary CD4 T cells including mouse and human primary T cells, stimulated with diverse TCR-triggering contexts, namely anti-CD3 and peptide-MHC complexes.

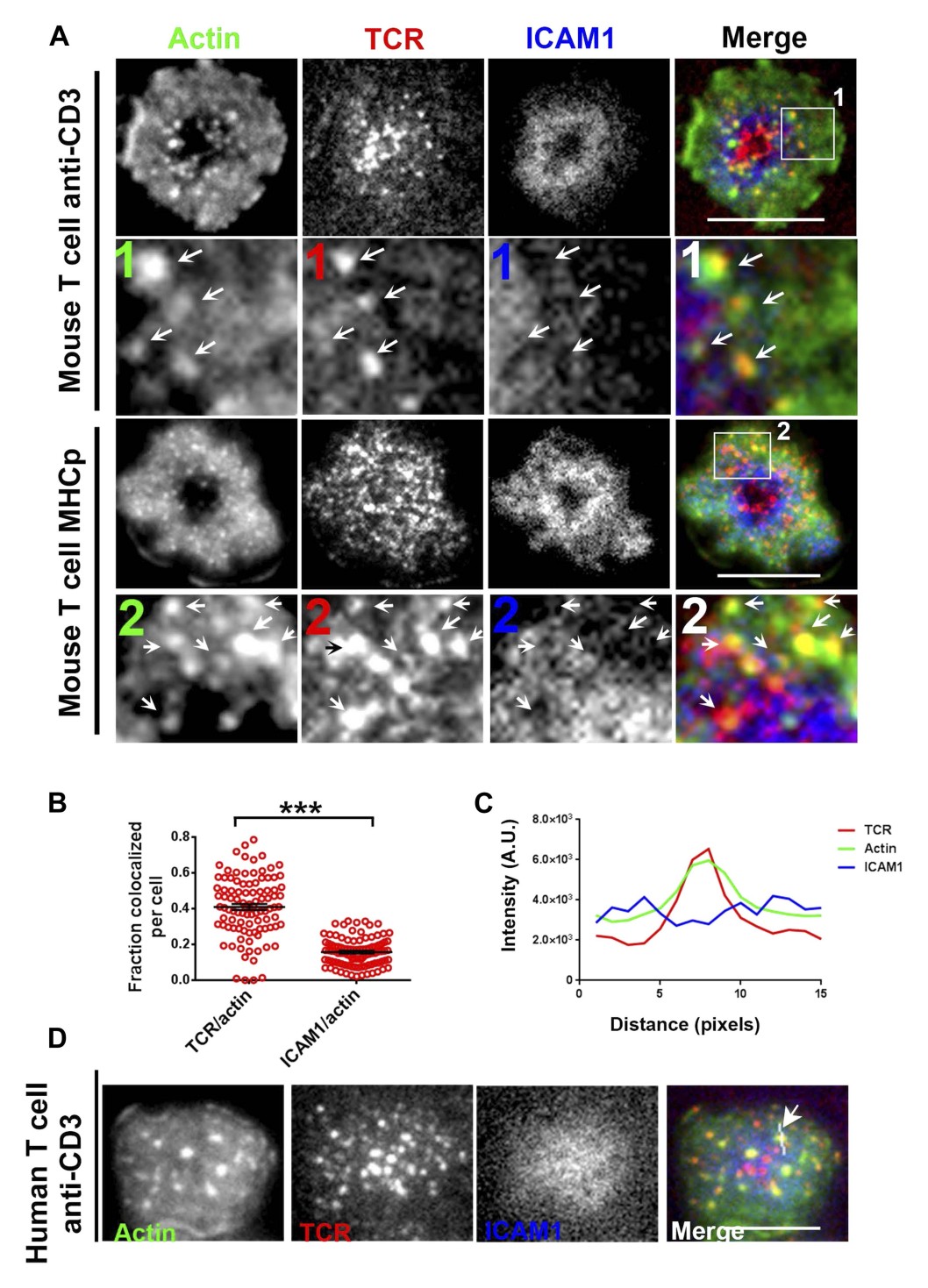

**Figure 2**. F-actin foci co-localize with TCR MC and not ICAM1. (**A**) Freshly isolated mouse AND CD4 T cells were incubated with lipid bilayer reconstituted with Alexa568 tagged anti-CD3 (TCR, red) and Alexa647-ICAM1 (blue), for 2 min at 37°C. Post incubation, cells were fixed and stained for F-actin using Alexa488-phalloidin (green), and imaged using TIRF microscopy. The region marked 1 in the 'merge' panel is magnified to clearly show the co-localization of actin foci with TCR-containing MCs. Lower panels: AND mouse CD4 T cell blasts exhibit F-actin enrichment at TCR MCs sites. AND mouse CD4 T cell blasts were labeled with Alexa568-H57 Fab (TCR), and incubated with bilayer reconstituted with MHCp and Alexa405-ICAM1 for 2 min at 37°C, fixed and stained for F-actin. Region marked 2 in the 'merge' image is further magnified to show the overlap between TCR and F-actin (arrows). *Figure 2. continued on next page*

*Figure 2. Continued*

The insets 1 and 2 in both MHC-activated and anti-CD3-activated mouse T cells are contrasted differently from the original 'merge' image to highlight the TCR and actin distribution. (**B**) Quantitation of the fraction of TCR or ICAM1 localized with F-actin foci. AND CD4 T blasts were incubated with antigen containing bilayer for 2 min, as described above and the images acquired were processed for colocalization assessment as described in 'Materials and methods' section. Each point represents fraction of total synaptic TCR or ICAM1 associated with F-actin foci in a single cell. n1 = 99, n2 = 100. p < 0.0001. (**C, D**) Freshly isolated human peripheral blood CD4 T cells were incubated with bilayer containing Alexa568 tagged anti-CD3 (TCR), Cy5-ICAM1 at 37°C, fixed and stained with Alexa-488 phalloidin and subsequently imaged using TIRF microscopy. The image shows a freshly attached cell to the bilayer. The line marked by the arrow in the 'merge' panel shows the line-scan profile plotted in (**C**), where relative intensities of ICAM1 and actin with single TCR MC across the pixels marked in the 'merge' image are shown. Scale bar, 5 μm.

The following figure supplements are available for figure 2:

**Figure supplement 1**. Test for co-localization of foci and MCs.

**Figure supplement 2**. Loss of WASP in Human CD4 T cells, and its impact on foci induction.

**Figure supplement 3**. (**A**) Increased phosphorylation of HS1 in response to TCR activation.

**Figure supplement 4**. Lack of F-actin foci in the cSMAC of primary T cells and the Jurkat T cell line.

**Figure supplement 5**. Association of TCR MC and F-actin foci in live T cell.

Since a fraction of TCR MC did not localize with foci, we sought to examine this population. High magnification microscopy of T cell synapses formed by mouse CD4 T cells on anti-CD3 and ICAM1 SLB provided insight into both synaptic and non-synaptic F-actin organization (*Figure 2—figure supplement 4*, *Videos 1–2*). As expected (*Stinchcombe et al., 2006*), the F-actin signal progressively decreases toward the synapse center where central TCR MC are devoid of F-actin foci (*Figure 2—figure supplement 4*, asterisks in the 'Merge' panel). To eliminate the possibility that the lack of foci on TCR MC in the nascent cSMAC is a consequence of cell fixation methodology, we examined foci in live T cells by transfecting them with LifeAct-GFP, a peptide construct that selectively labels F-actin (*Riedl et al., 2008*). Since primary mouse T cells showed poor survival after transfection, we used human primary CD4 T cells that exhibit higher viability (*Chicaybam et al., 2013*). In live cell synapses, LifeAct-GFP was observed to form foci on TCR MCs, which continued to co-migrate until the delivery of MCs to the cSMAC (*Figure 2—figure supplement 5*, *Video 3*). Analysis of kymographs revealed that the average TCR MC speed was 5.66 ± 2.2 μm/min, average Foci speed was 6.63 ± 3.1 μm/min (p = 0.21, ns). The TCR signals persisted in the cSMAC, whereas the GFP foci were extinguished, consistent with continual nucleation of F-actin at the MC site (*Figure 1A*) until it reaches the cSMAC, where TCR signals are known to be terminated (*Vardhana et al., 2010*; *Choudhuri et al., 2014*).

We also note that the Jurkat T cell line stimulated using a similar anti-CD3 and ICAM1 activation system displayed only random co-localization of TCR MC and F-actin (9.0% ± 0.09 TCR colocalized with F-actin, n = 47, *Figure 2—figure supplement 4*, right panels, *Video 4*). This suggests that a distinct F-actin organization may exist in this commonly used leukemia derived cells line.

Since F-actin foci are associated with TCR MC sites in pSMAC and dSMAC zones, we sought to investigate the requirement of TCR signals for their formation. AND T cells activated using ICAM1 alone displayed some interface F-actin (*Figure 3A*, top images); however, a significant induction of F-actin foci only occurred when AND TCR transgenic T cells were incubated with both ICAM1 and agonist peptide-MHC (*Figure 3A*, top graph). A similar requirement of TCR triggering for F-actin foci formation was observed in primary human CD4 T cells (*Figure 3A*, bottom graph), although it did not scale with the dose of TCR agonist; in T cells incubated with low or high concentrations of anti-CD3 antibody, we did not observe a corresponding increase in F-actin foci (*Figure 3—figure supplement 1*). These results show that TCR activation triggers robust foci formation above a threshold level of

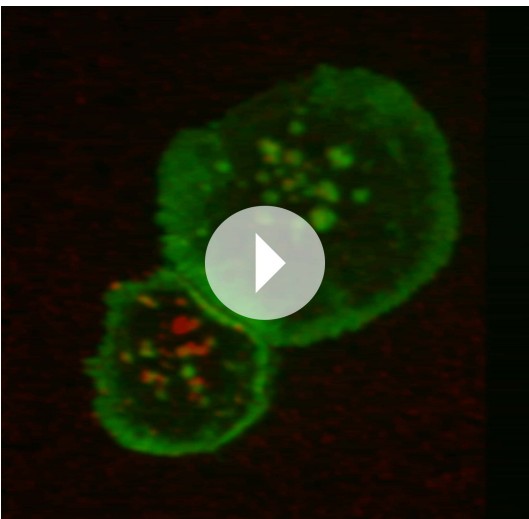

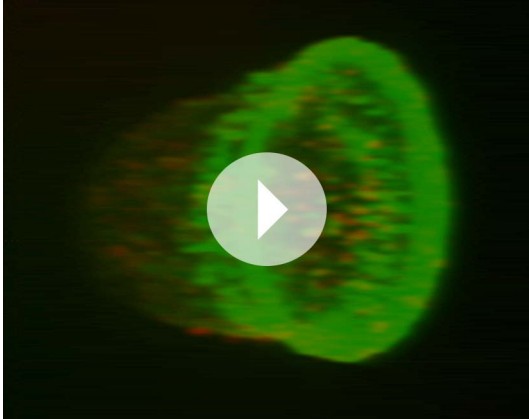

**Video 1.** AND CD4 T cells were incubated with bilayer containing ICAM1 and Alexa568 tagged anti-CD3 (red) for 2 min, fixed and stained with Alexa488-phalloidin (green). Cells were then visualized using spinning disc confocal microscopy. Video shows 3D reconstruction of images acquired.

**Video 2.** AND CD4 T cells were labeled with Alexa568-H57 Fab (red), were activated on bilayer containing MHCp and ICAM1 for 2 min, then fixed and stained with Alexa488-phalloidin (green), and eventually imaged using spinning disc confocal microscopy. Video shows 3D reconstruction of images acquired.

TCR engagement, in the presence of a constant level of ICAM1 mediated adhesion. The residual foci intensity observed in T cells in the ICAM1 alone case (*Figure 3A* graph) is primarily due to fluctuations in lamellipodial F-actin detected as foci by our analysis method (*Figure 1—figure supplement 1,2*).

To further investigate the role of TCR signaling in foci formation, we examined the role of Src family kinases (SFK) signaling. TCR signaling proceeds by accumulation of active Lck, as indicated by phosphorylation of the activation loop, in MCs (*Campi et al., 2005*). Consistent with this, phospho-SFK localized to F-actin foci associated MC (*Figure 3—figure supplement 2*). Furthermore, inhibition of SFK phosphorylation using PP2 (*Varma et al., 2006*) significantly reduced the number of F-actin foci (*Figure 3B*), indicating that TCR-driven SFK signaling is required for their

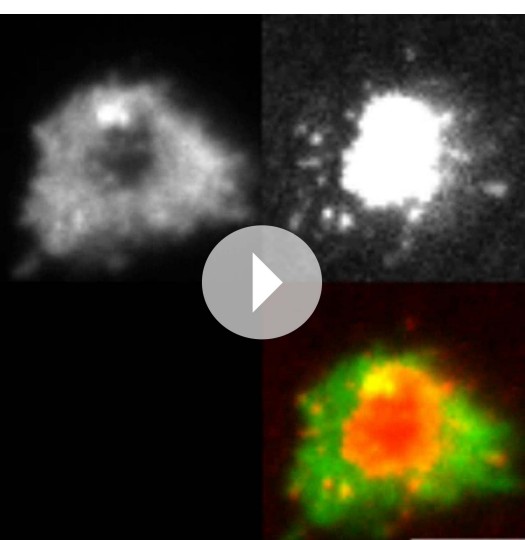

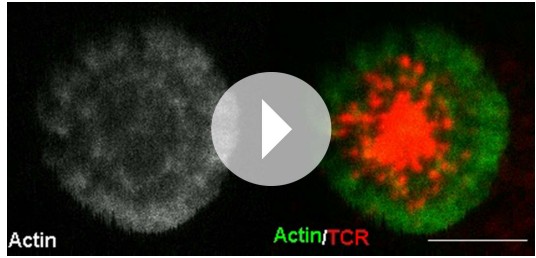

**Video 3.** Human CD4 T cells were transfected with Lifeact-GFP (pseudocolored green) and were incubated with bilayer containing ICAM1 and Alexa568 tagged anti-CD3 (pseudocolored red). The video was acquired with 5 s intervals between frames. The play rate is 80 times faster than the acquisition rate. Scale bar, 5 µm.

**Video 4.** Jurkat T cells were transfected with Lifeact-GFP (pseudocolored green) and incubated with bilayer containing ICAM1 and Alexa568 tagged anti-CD3 (pseudocolored red). The video was acquired with 15 s interval between frames. The play rate is 33 times faster than the acquisition rate. Scale bar, 10 µm.

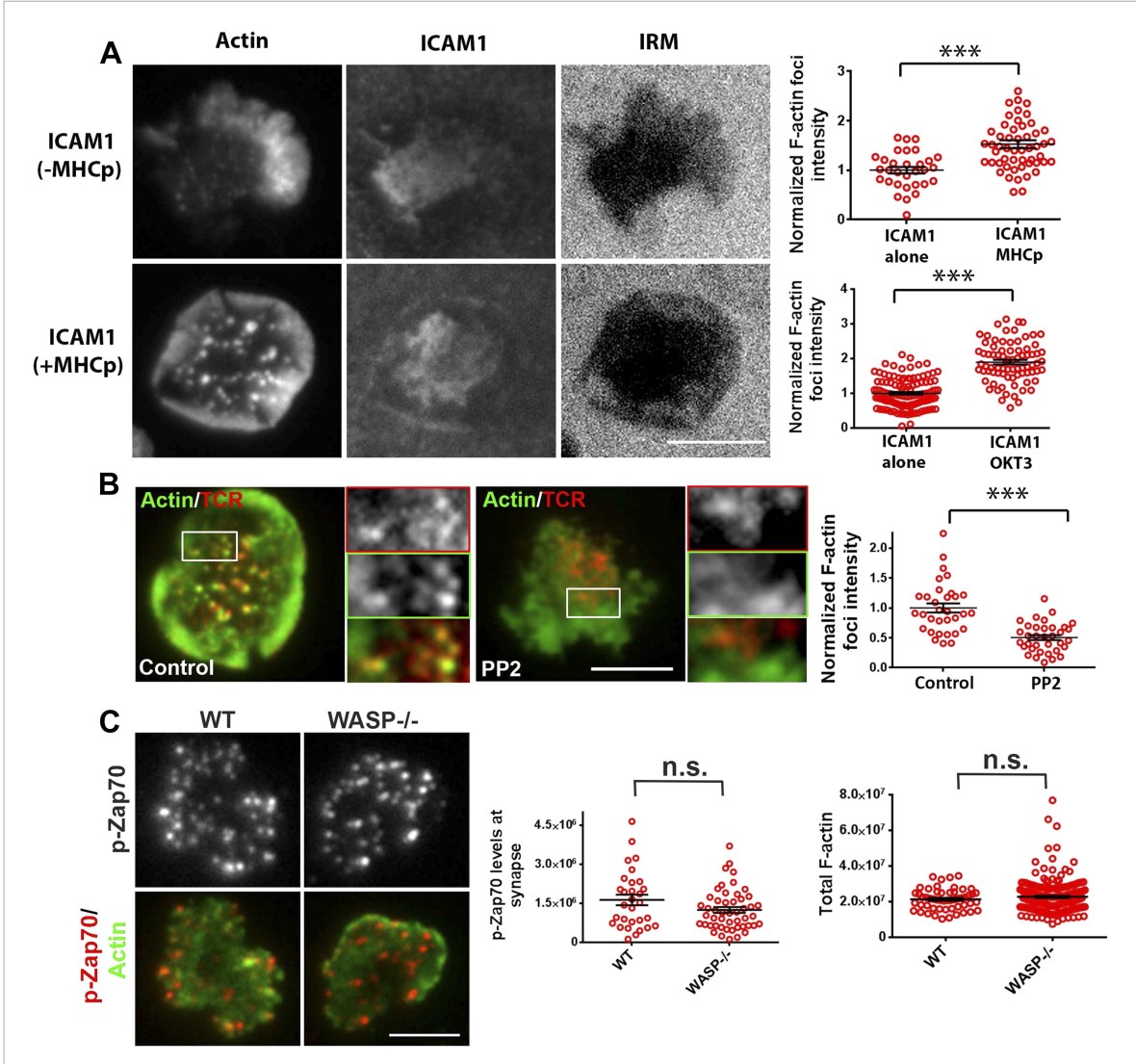

**Figure 3**. Relationship of F-actin foci to TCR engagement and TCR proximal signaling. (**A**) AND mouse CD4 T cell blasts were incubated with lipid bilayer containing Alexa405-ICAM1 and MHCp, or bilayer containing Alexa405-ICAM1 alone (ICAM1) at 37°C for 2 min. The cells were fixed and stained with Alexa488-phalloidin (Actin) and imaged using TIRF microscopy. The images in the right panel are reflection (interference reflection microscopy – IRM) images in the two conditions, showing cell-bilayer contact area. Actin images were further high-pass filtered using a rank-filter based subtraction method (see 'Materials and methods') to reveal spatially localized actin features within each cell, under both incubation conditions. The graph (top right) shows average intensity of actin features (F-actin foci) per cell. n1 = 31, n2 = 53, p < 0.0001. The bottom graph shows foci induction in human CD4 T cells. Primary human CD4 T cells were incubated with bilayer containing Alexa405-ICAM1 alone, or both Alexa405-ICAM1 and Alexa568 tagged anti-CD3 at 37°C for 2 min, and were then fixed and stained for F-actin using Alexa488-phalloidin, for imaging using TIRF microscopy. The graph shows integrated intensity of actin spots per cell, each point on the graph represents a single cell. n1 = 117, n = 79, p < 0.0001 (**B**) Formation of TCR MC associated F-actin foci requires SFK signaling. AND T cell blasts were treated with PP2 for 10 min at 37°C and were labeled with Alexa568-H57 Fab. This was followed by further incubation with ICAM-1/MHCp containing bilayer at 37°C for 2 min, and fixation and staining for F-actin. The TIRF images show Actin (Alexa488-phaloidin, green) and TCR (red) distribution in DMSO treated (control, left) and PP2 treated (right) cells. Graph (far right) shows mean intensity of F-actin features in control and PP2 treated cells. n1 = 32, n2 = 35, p < 0.0001. (**C**) Phospho-Zap70 levels are normal in WASP deficient cells. Freshly isolated C57BL/6J WT or WASP–/– CD4 T cells were activated on bilayer containing anti-CD3 and ICAM1 for 2 min, fixed and stained with anti-phospho-Zap70 antibody (left graph). The top images show phospho-Zap70 intensity at the synapse, and the left graph shows integrated intensity of phospho-Zap70 per cell. n1 = 30, n2 = 49, p = 0.17. The WASP–/– cells display normal whole cell levels of TCR-induced F-actin, as marked by Alexa488-phalloidin staining (right graph). The cells were imaged under wide-field illumination settings – each point represents total F-actin intensity per cell. n1 = 53, n2 = 212, p = 0.583.

The following figure supplements are available for figure 3:

**Figure supplement 1**. Effect of TCR-ligand concentration on F-actin foci formation.

*Figure 3. continued on next page*

*Figure 3. Continued*

**Figure supplement 2**. Localization of phosphorylated form of Src family kinases (SFK) at TCR MC/F-actin foci.

formation. We conclude that F-actin foci correspond to sites of early TCR signaling and require active SFKs.

The next step in TCR proximal signaling is the SFK mediated recruitment of Zap70 to the TCR cytoplasmic domains and the phosphorylation of Zap70 by SFK on sites including Y319 in the interdomain linker. TCR-induced Zap70 Y319 phosphorylation in MC and overall F-actin levels at the synapse were comparable between primary WT and WASP−/− cells (*Figure 3C*). This result demonstrates that WASP is dispensable for TCR-proximal signaling including Zap-70 recruitment and activation.

## F-actin foci underlie WASP's function in calcium ion signaling

Having found that TCR MC formation and proximal signaling is independent of WASP, we investigated further TCR-distal signaling events that would lead to WASP-dependent calcium ion flux, as reported previously in human and mouse T cells (*Zhang et al., 2002*; *Calvez et al., 2011*). PLCγ1 is one of the key effector molecules regulating calcium ion flux downstream of TCR-activation, which mediates inositol-1,3,4-trisphosphate synthesis in the plasmamembrane, thereby facilitating calcium ion release from endoplasmic reticulum stores (*DeBell et al., 1992*; *Babich et al., 2012*). In addition, there is evidence that PLCγ1 can bind to F-actin (*Carrizosa et al., 2009*). We therefore reasoned that F-actin foci may locally enrich PLCγ1 at the TCR MC, supporting PLCγ1 interaction with TCR signalosome effectors such as Itk and LAT, thereby promoting its phosphorylation and activation (*Braiman et al., 2006*). We therefore assessed the role of WASP in PLCγ1 activation in WT and WASP−/− T cells. WASP−/− T cells exhibited significantly reduced phospho-PLCγ1 levels at the synapse (*Figure 4A*). The reduction in phospho-PLCγ1 levels was not due to a general decrease in total PLCγ1 levels in the WASP−/− T cells, as the levels of total cellular PLCγ1 were comparable in WT and WASP −/− whole T cell lysates (*Figure 4B*). Thus, a loss of WASP-driven F-actin foci is correlated with impairment of PLCγ1 activation at the synapse.

We next asked whether the well-established role of WASP in T cell calcium ion signaling pathway could be accounted for by F-actin foci, or by other pathways involving WASP-dependent protein–protein interactions. WASP is a scaffolding protein and interacts with a variety of proteins at MCs (*Thrasher and Burns, 2010*). To distinguish between two possible roles of WASP, activating Arp2/3 complex to generate F-actin foci, or acting as a direct molecular scaffold, in the regulation of PLCγ1 activation at the synapse, we overexpressed mutant forms of human WASP, including an Arp2/3-activation deficient form (WASP Δ473–480 or WASPΔC) and SFK scaffolding deficient form (WASPY291F) in human CD4 T cells. WASP binds with Arp2/3 via its C-terminal verprolin-homology, central, acidic (VCA) domain, while a stretch of hydrophobic residues in the central domain is reported to facilitate Arp2/3 activation without affecting the binding of Arp2/3 to the acidic domain (*Machesky and Insall, 1998*; *Panchal et al., 2003*). WASPΔC lacks this Arp2/3 activation sequence within the C-region (*Kato et al., 1999*) (*Figure 4C*, schematic). Indeed, T cells expressing GFP-WASPΔC showed normal levels of Arp2/3 complex recruitment (data not shown), as well as total F-actin at the synapse, and synaptic recruitment of WASPΔC was comparable to WT WASP (*Figure 4C,D*). However, WASPΔC expressing cells were severely defective in F-actin foci generation, as well as in PLCγ1 phosphorylation at the synapse (*Figure 4C,D*). This result indicates that WASP-dependent Arp2/3 activation is essential for foci genesis and activation of calcium ion signaling effector PLCγ1 at the synapse. Overexpression of WASP Y291F (*Dovas et al., 2009*), a mutant form defective in phosphorylation dependent interaction with SH2 domain of SFKs (*Padrick and Rosen, 2010*), had no effect on foci formation or PLCγ1 activation. This result suggests that WASP's scaffolding of SFKs via Y291 does not play a role in foci formation or PLCγ1 phosphorylation (*Figure 4D*). Together these results demonstrate a critical requirement of WASP's Arp2/3 activation and foci nucleation ability for PLCγ1 activation. Thus, decoupling of WASP's interactions in MC from the ability to activate the Arp2/3 complex, but not SFKs, results in a strong disruption of F-actin foci.

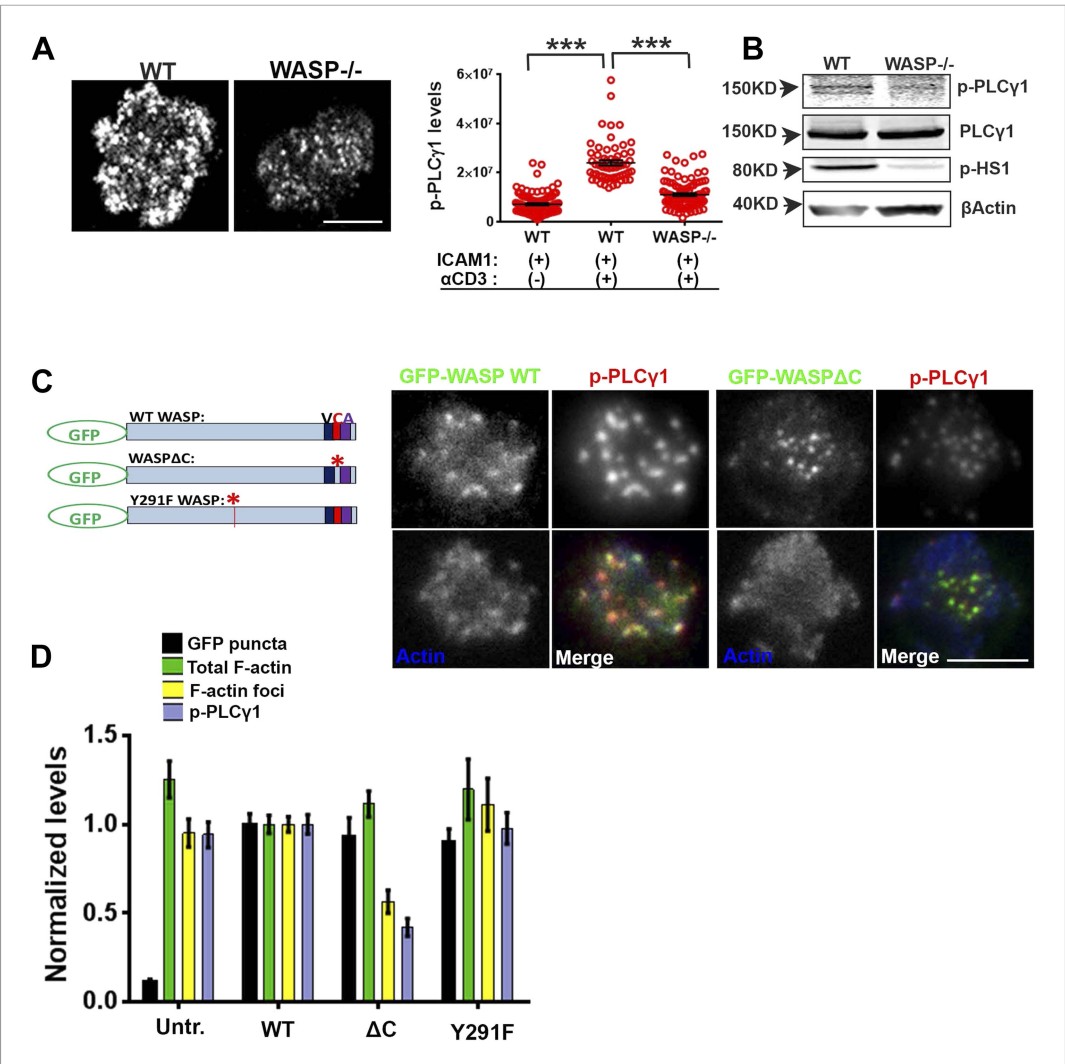

**Figure 4**. WASP regulates calcium ion signaling via generation of F-actin foci. (**A**) WASP deficient T cells exhibit impaired TCR-induced PLCγ1-Y783 phosphorylation. CD4 T cells freshly isolated from C57BL/6J WT or *Was*−/− mice were activated with ICAM1 alone, or both ICAM1 and anti-CD3 for 2 min. Cells were fixed and immunostained for phospho-PLCγ1 (image not shown), and visualized using confocal microscopy. The images show phospho-PLCγ1 staining in the bottom section (synapse plane) of WT (left) or WASP−/− (right) cells. The graph on the left shows phospho-PLCγ1 levels in the synapse planes in the cells in WASP deficiency background. n1 = 104, n2 = 60, n3 = 94, *p < 0.0001*. (**B**) Assessment of total cellular PLCγ1 phosphorylation in WASP−/− T cells using western blot. Freshly isolated WT or WASP−/− CD4 T cells were incubated with anti-CD3/CD28 beads for 5 min, lysed, and the lysates were analyzed using western blotting. Note that, in WASP−/− T cells TCR-induced PLCγ1 phosphorylation is defective, while total PLCγ1 is comparable to the WT cells. HS1 phosphorylation was included as a control that exhibits diminished phosphorylation in WASP−/− T cells. These experiments were repeated twice with similar results. (**C**) Arp2/3 activation by WASP is essential for F-actin foci generation and optimal phospho-PLCγ1 at the synapse. Human CD4 T cells were transfected with GFP-WASP (WT), GFP-WASPΔC, or GFP-WASP291F (shown in the schematic on the left) for 16 hr and were then incubated with anti-CD3/ICAM1-reconstituted bilayers for 2 min, fixed and stained with Alexa568-phalloidin and anti-phospho-PLCγ1 antibody. The images show the GFP (green), F-actin (blue) and phospho-PLCγ1 (red) distribution at the synapse for WT (left) and WASPΔC (right) T cells. (**D**) The graph shows levels of GFP-tagged constructs at the synapse, analyzed and obtained via 50% rank filtering of images shown in (**D**), as described in 'Materials and methods', as well as the quantitation of total synaptic F-actin, foci and phospho-PLCγ1 in the same cells, normalized to mean values obtained for WT cells. n1 = 23, n2 = 27, n3 = 33, n4 = 27 p *values*, p < 0.0001 between WT and WASPΔC cells for foci and phospho-PLCγ1 levels, and between untransfected and cells expressing GFP tagged constructs for 'GFP puncta'. For all other comparisons, p > 0.05.

## Arp2/3 complex polymerizes F-actin foci at TCR MCs

Next, we more directly addressed the role of Arp2/3 complex, which is known to nucleate actin filaments de novo downstream of WASP (*Dominguez, 2010*), in foci induction. A previous study utilizing shRNA-mediated silencing of the Arp2/3 complex subunits in Jurkat T cells had demonstrated that actin polymerization at the synapse can occur independent of the Arp2/3 complex (*Gomez et al., 2007*). However, the high-resolution subsynaptic distribution of Arp2/3 complex at the primary T cell synapse remains unknown. We first assessed the association of Arp2/3 complex with TCR MCs. A significant fraction of endogenous Arp2/3 complex is recruited at the TCR MC sites (*Figure 5A*, *Figure 5—figure supplement 1A*). In T cells treated with CK666, a small molecule inhibitor affecting conformational change in Arp2/3 complex and thus actin nucleation (*Nolen et al., 2009*), the association of both Arpc2 and 3 with TCR MCs was reduced (*Figure 5A*, *Figure 5—figure supplement 1A*). The same treatment conditions also led to loss of F-actin foci (*Figure 5B*). As a control, CK689, an inactive isomer of CK666 (*Figure 5—figure supplement 1B*), and the inhibitor for another class of actin nucleation factors, formins (*Rizvi et al., 2009*), or inhibition of myosin II using blebbistatin (*Straight et al., 2003*), did not inhibit F-actin foci (*Figure 5B*, *Figure 5—figure supplement 1C*), indicating that these foci specifically require Arp2/3 complex for nucleation, and myosin-mediated actin rearrangement is not required for their formation.

Since CK666 treatment can rapidly eliminate foci, we utilized CK666 to probe the kinetics of foci formation at individual TCR MCs. Given the small size and large lamellar actin background at the primary T cell synapse as well as highly mobile nature of MCs, this analysis would be extremely challenging to perform in the bilayer system. Therefore, we activated T cells on micron-scale anti-CD3 dots adsorbed on a glass surface where the remaining space was coated with ICAM1 in an attempt to restrict TCR signaling to a known, micron scale location in an artificial immunological synapse. Due to their aforementioned performance in time-lapse studies after transfection, human CD4 T cells were utilized for LifeAct-GFP transfection and live imaging on anti-CD3 microdots. Incubation with the patterned surface triggered rapid actin polymerization at the anti-CD3 microdot, reflected in LifeAct-GFP accumulation (*Figure 5C–E*). This enrichment of LifeAct at the microdot was not due to the topography of the cell contact interface (*Owen et al., 2013*), since a global membrane labeling dye, DiI, and integrin LFA1 were not enriched at these sites (*Figure 5—figure supplement 2A,B*). Within 5 s of CK666 treatment, pre-existing LifeAct enrichment was lost from the microdot site (*Figure 5C–E Video 5*) and fresh microdot contact events failed to elicit LifeAct-GFP enrichment in the presence of CK666 (*Video 5*). These data confirm that the TCR engagement sites dictate Arp2/3 complex-dependent F-actin foci formation at the synapse, de novo. Furthermore, ligand microdots can spatially pattern F-actin foci by predefining the sites of TCR engagement, and thus provide a suitable platform for spatially isolating TCR dependent mechanisms.

## F-actin foci and calcium ion signaling at the synapse

We have shown above that WASP's foci forming ability is required for PLCγ1 activation (*Figure 4B*). In order to test whether the independent elimination of foci by perturbation of Arp2/3 complex would phenocopy this effect, we chose to deplete foci using CK666 and monitor the consequences of this treatment on early TCR signaling, PLCγ1 activation and subsequent calcium ion signaling. CK666 treatment, which resulted in severe inhibition of F-actin foci (*Figure 5B*, *Figure 6—figure supplement 1B*), did not diminish TCR-proximal activation events. While fewer T cells formed synapses after CK666 treatment (37% reduction), upon adhesion, F-actin dependent processes including formation of TCR MC and exclusion of CD45 phosphatase from MCs (*Campi et al., 2005*; *Varma et al., 2006*) (*Figure 6A*), and the phosphorylation of Zap70 on Y319 (*Figure 6B*) proceeded normally in CK666-treated cells. In addition, other TCR-activation dependent signaling events leading to calcium flux, such as the phosphorylation of LAT on Y171 and SLP76 on Y145 were not altered in CK666-treated cells in comparison to control treated cells (data not shown). However, a severe defect was observed in the levels of TCR-induced PLCγ1 phosphorylation on Y783 after CK666 treatment (*Figure 6C,D*), where a drop was observed in both the number and intensity of phospho-PLCγ1 clusters at the synapse (*Figure 6C*). Furthermore, while phospho-PLCγ1 localized with F-actin foci in control cells, treatment with CK666 led to loss of foci-associated phospho-PLCγ1 (*Figure 6E*, *Figure 6—figure supplement 1A*). Upon attenuation of F-actin foci, total PLCγ1 levels were also reduced at the synapse, matching the decline in phospho-PLCγ1 levels (*Figure 6F*) and indicating that the impaired

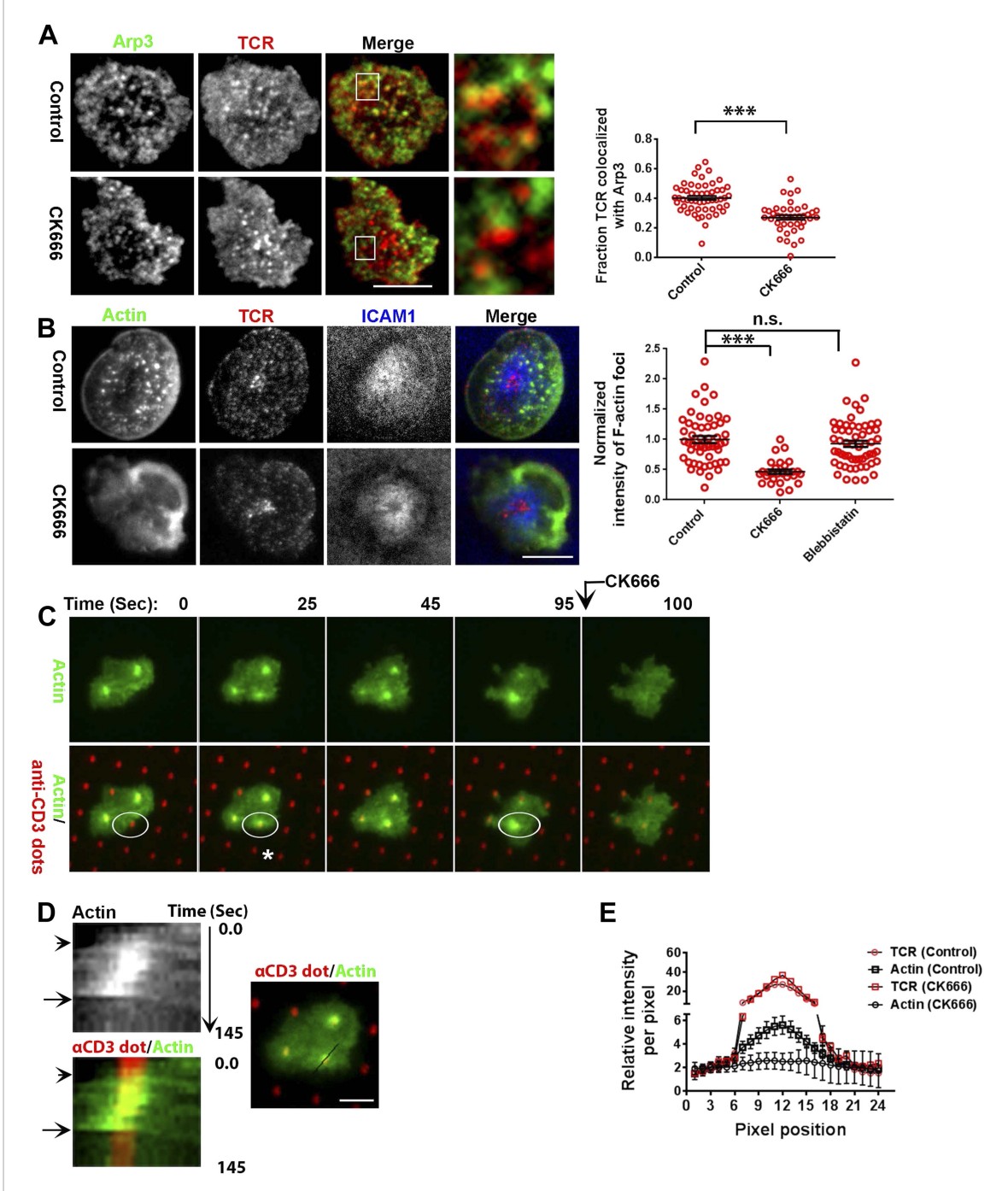

**Figure 5**. F-actin foci require activity of the Arp2/3 complex. (**A**) Arp2/3 complex is localized at TCR MCs. DMSO (Control, top) or CK666 treated-(bottom) AND mouse T cell blasts were labeled with Alexa568-H57 Fab and were incubated with bilayer containing ICAM1 and MHCp for 2 min at 37°C, and then fixed and immunostained with anti-Arp3 antibody. The TIRF images show relative distribution of Arp3 (left, green) and TCR (middle, red) in the TIRF plane. Note that while Arp3 distribution overlaps with MCs in control cells, MCs are significantly devoid of Arp3 in CK666-treated cells (areas from 'merge' panels further magnified in insets). The graph (far right) shows the fraction co-localization of total synaptic TCR with Arp3. n1 = 54, n2 = 41, p < *0.0001*. In this experiment, the mean intensity of Arp3 in synapse was 3.46 ($\pm$0.022) × $10^6$ for control cells, and 2.38 ($\pm$0.015) × $10^6$ for CK666 treated cells (data not shown), and it was repeated twice with similar results. (**B**) TCR MCs associated F-actin enrichment is generated by Arp2/3 complex activation. AND T cell blasts were treated with DMSO alone (control), or 100 µM CK666, or 100 µM blebbistatin for 10 min at 37°C, were labeled with Alexa568-H57 Fab (TCR, red), and incubated with Alexa405-ICAM1 (ICAM1, blue) and MHCp-containing bilayer in the presence of the specific inhibitors at 37°C for 2 min. Cells were subsequently fixed, stained with Alexa488-phalloidin and imaged using TIRF microscopy. Note that there is a substantial reduction in number of actin foci in the cells treated with CK666. The graph (far right) shows the quantitation of F-actin foci intensity in the cells in inhibitor treatment backgrounds. n1 = 52,
*Figure 5. continued on next page*

*Figure 5. Continued*

n2 = 28, n3 = 56. p1 < 0.0001, p2 = 0.36. (**C**) Immobilized anti-CD3 microdots induce localized actin polymerization that is dependent on Arp2/3 complex. Human CD4 T cells were transfected with LifeAct-GFP plasmid (green), incubated with glass substrate coated with ICAM1 and printed with Alexa647 tagged anti-CD3 microdots (red), and subsequently imaged live. The images represent snapshots taken from a time-lapse sequence, at the indicated time points, '0 s' represents onset of the video sequence acquisition. Note that immediately upon contacting the microdot, cells display localized actin polymerization (marked circle and *). CK666 treatment (arrow after 95 s panel) leads to immediate loss of F-actin from the foci. (**D**) Kymograph of F-actin events (upper left image) with anti-CD3 dot (bottom left image) during contact formation with the microdot prior to and during CK666 treatment. Arrowhead indicates the time of contact of the cell with anti-CD3 dot, and arrow indicates start of CK666 treatment. The line marked on the color combined image (right) shows the region of the cell used to create kymograph for the indicated time duration. (**E**) Enrichment of LifeAct (Actin, black) at the anti-CD3 microdot (TCR, red) in control and CK666-treated fixed cells. The graph shows the mean intensity per pixel obtained using the line-scan profile of LifeAct (actin) and anti-CD3 (TCR) across 31 microdots from 15 cells. Intensity profiles across identical lines for each microdot were obtained for both anti-CD3 and LifeAct; these values were normalized to the lowest pixel intensity for each line-profile, and their mean ± SEM values are plotted in the graph.

The following figure supplements are available for figure 5:

**Figure supplement 1**. Arp2/3 and formin's role in F-actin foci.

**Figure supplement 2**. F-actin foci on microdots are not enriched in phospholipid membranes.

levels of phospho-PLCγ1 at the whole cell level may be accounted for by the reduction in PLCγ1 enrichment at the TCR signalosome. When assaying signaling events downstream of PLCγ1 activation, such as TCR-induced calcium ion elevation in the cytoplasm and nuclear mobilization of NFAT1, CK666-treated cells showed significant reduction in these processes (*Figure 6G,H*). Although a reduction in lamellipodial and lamellar actin was observed in CK666 treated cells (mean total F-actin 42% reduced), this reduction was relatively lower than foci depletion (mean 67% reduced) or under global F-actin depletion (CytoD, mean F-actin 90% reduced), and did not affect early TCR signaling (*Figure 6—figure supplement 1B*). We thus conclude that WASP and Arp2/3 dependent foci play a non-redundant role in PLCγ1 activation and calcium ion signaling at the synapse. In line with this result, CK666-treated cells exhibited a loss of phospho-HS1, as well as HS1, at the synapse (*Figure 6—figure supplement 1C*), similar to that observed in WASP−/− T cells (*Figure 1E*). Similar to the TCR-distal signaling defects observed in CK666 treated mouse CD4 T cells described above, CK666-treated human CD4 T cells also displayed a reduction in phospho-PLCγ1, but not in phospho-Zap70 (*Figure 6—figure supplement 2*), indicating that foci-dependent PLCγ1 activation cascade is conserved in T cells from diverse origins.

To directly visualize the involvement of F-actin foci in PLCγ1 recruitment and stabilization at the TCR signalosome in live cells, we transiently overexpressed PLCγ1-YFP in human CD4 T cells, and examined its distribution at the synapse on SLB presenting anti-CD3 and ICAM1 in real time. PLCγ1 localized with TCR microclusters in live cell synapse, and this distribution was lost upon treatment of cells with CK666 (*Figure 6—figure supplement 3*). As a control, the association of human Zap70-GFP with MC was maintained after CK666 treatment (*Figure 6—figure supplement 3*). However, due the mobile nature of MC, it was challenging to follow individual cluster before and after CK666 treatment in this setting. We thus chose to utilize anti-CD3 microdots to activate human T cells. Human CD4 T cells expressing PLCγ1-YFP were incubated with micron-size immobile anti-CD3 dots, and visualized using live cell imaging immediately after cell attachment. Similar to F-actin foci assembly at the microdot, PLCγ1 enrichment was also visible at the microdot sites. CK666 treatment of cells led to a reduction of PLCγ1 enrichment at microdots (*Figure 6I,J*). These data show that TCR MC associated F-actin foci assist in sustained accumulation of PLCγ1 at the TCR MC.

## F-actin foci at T cell-APC conjugate interface

While SLB and solid-phase adsorbed antigen presentation systems allowed for superior optical resolution of the synapse, it is important to further test the validity of predictions from these reductionist models in more physiological T cell-APC interactions. Thus we utilized an activated cultured endothelial cells (EC) based planar APC system, where a flat synaptic interface is formed that

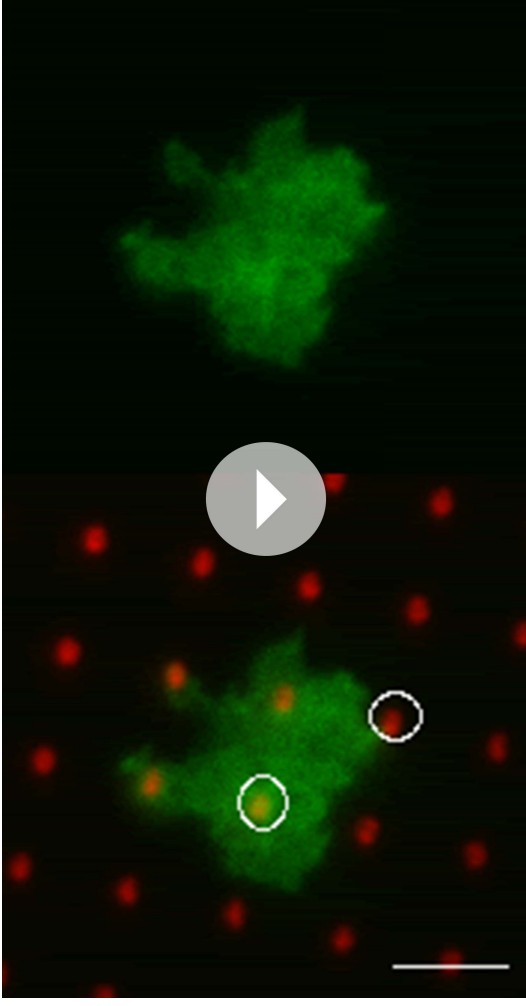

**Video 5.** Human CD4 T cells were transfected with LifeAct-GFP plasmid (green) and incubated with ICAM1-coated glass substrate with 1 μm anti-CD3 printed dots (red) on stage at 37°C and imaged live. The play rate is 25 times faster than the acquisition rate. In all of the above images and videos, scale bar is 5 μm, unless otherwise noted.

is ideal for *en-face* visualization (*Sage et al., 2012*). Using this system, F-actin and HS1 rich 'invadopod-like-protrusions' (ILPs) are observed at TCR MC-like features in the periphery of the synapse (*Sage et al., 2012*). We found that ILPs were associated with the Y397 phosphorylated form of HS1, similar to F-actin foci (*Figure 7A*). Furthermore, when pre-formed T cell-EC conjugates were treated with CK666, there was concomitant loss of ILPs and phospho-HS1, indicating that these structures also require Arp2/3-dependent continuous polymerization of F-actin, and support local phospho-HS1 levels (*Figure 7—figure supplement 1A–C*). Since CK666-treatment also led to a reduction in total synaptic F-actin, which could be a consequence of inhibition of the EC cytoskeleton (*Figure 7—figure supplement 1A–C*), we utilized specific shRNA-mediated reduction in WASP levels in T cells, and examined foci (ILPs), total F-actin and phospho-HS1 levels at the T cell-EC conjugate interface (*Figure 7A,B*). WASP silenced T cells display selective loss of both ILPs as well as phospho-HS1, while maintaining total F-actin levels (*Figure 7A–C*). Therefore, the F-actin foci that we have primarily been characterized using SLB in this study, are the counterparts of ILPs in the cell conjugate system, and are necessary for efficient T cell activation. Thus, F-actin foci may represent a cytoskeletal module that functions to optimally support TCR distal signaling (*Figure 7C*) in diverse antigen presenting contexts.

## Discussion

### A unified view of the basic TCR signaling unit

Compartmentalization of signaling proteins is a well-established strategy for spatial regulation of signaling in mammalian cells. At the immunological synapse, the compartmentalization of TCR signalosome proteins is regulated for optimal function (*Fooksman et al., 2010*). It is therefore surprising that cytoskeletal behavior is routinely measured as an all-or-none response of total polymerized F-actin, effectively ignoring different F-actin network subpopulations that could have distinct contributions to specific steps of the TCR signaling process. The WASP-dependent F-actin microarchitecture at the TCR MC that we report here is an important example. These F-actin 'foci' facilitate specific downstream TCR signaling steps, including PLCγ1 activation and calcium ion elevation, but are not required for TCR MC formation and other proximal signaling events such as Zap70 phosphorylation. Our study has shown that, when F-actin foci are depleted by WASP deficiency or CK666 treatment, the remaining lamellipodial and lamellar F-actin networks are not able to support optimal PLCγ1 activation, indicating that actin foci are indeed necessary for this later signaling step. Furthermore, we found that the molecular machinery associated with 'flat' TCR MC formed on planar substrates is the same as that in the distinctly three-dimensional ILP structures formed in T cell interfaces with antigen presenting endothelial cells. Others have noted the three dimensional nature of T-APC synapses that appear to have a classical immunological synapse/SMAC architecture

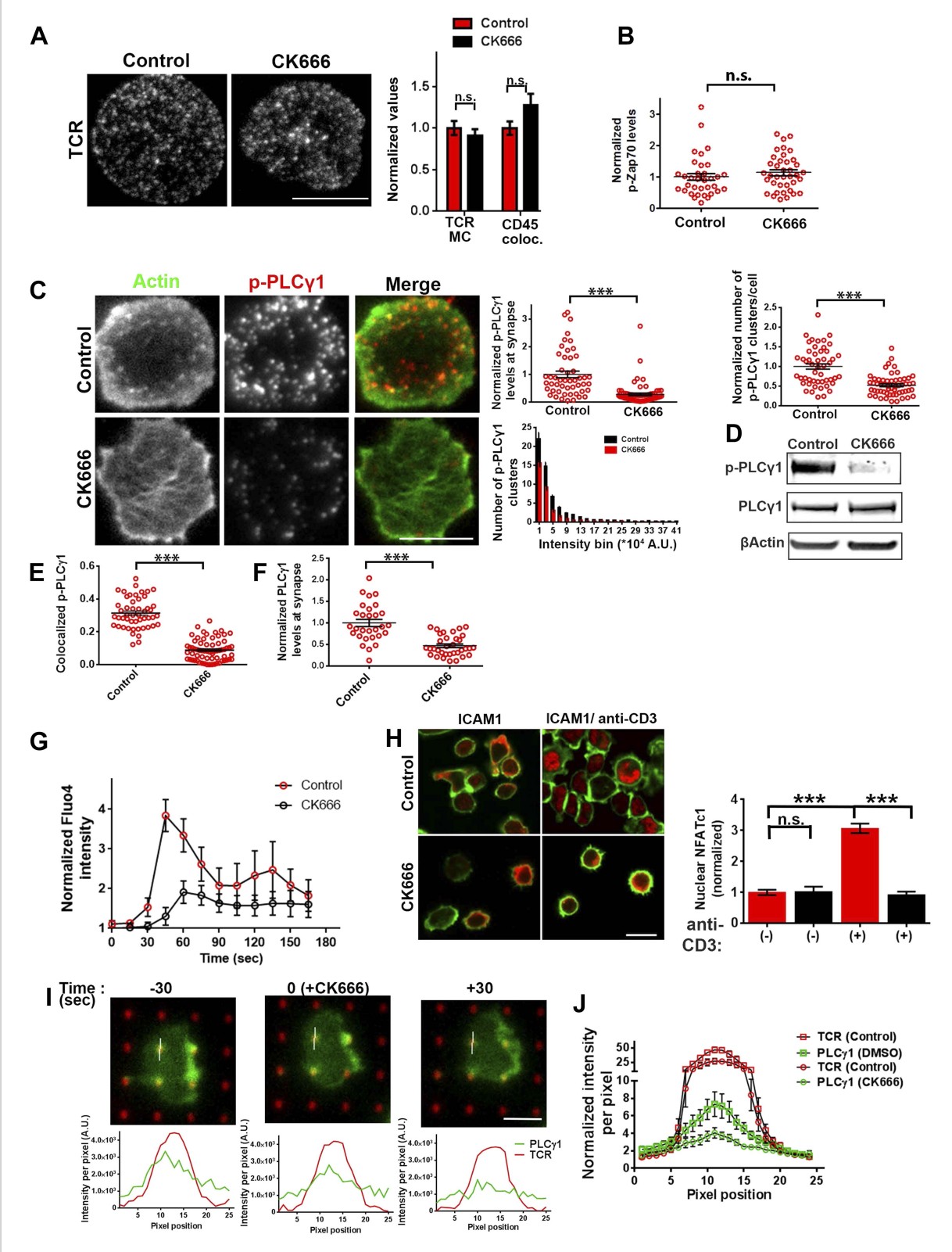

**Figure 6.** Arp2/3 inhibition leads to defective TCR-distal signaling. (**A**) Formation of TCR MCs or CD45 exclusion does not require F-actin foci. AND CD4 T cell blasts labeled with Alexa568-H57 Fab (to assess TCR clustering), or Alexa488-CD45 Fab (for CD45 exclusion) at 4ºC, were then incubated with bilayers containing ICAM1 and MHCp for 2 min, fixed (to assess TCR clustering) or visualized live (for CD45 exclusion) using TIRF microscopy. The images were

*Figure 6. continued on next page*

*Figure 6. Continued*

processed to assess TCR clustering, or CD45 and TCR colocalization using rank filter based filtering, as described in 'Materials and methods' section. For the bars showing TCR cluster intensities per cell, n1 = 45, n2 = 28, p = 0.6; for CD45 co-localization, n1 = 26, n2 = 13, p = 0.09. (**B**) Phosphorylation of TCR-proximal molecule Zap70 is not reduced in the cells treated with CK666. T cells were treated with DMSO or CK666 for 10 min, then incubated with surface containing ICAM1/anti-CD3 for 3 min, and processed for Y319-phospho-Zap70 and imaged using TIRF. The images were quantified to obtain the synaptic levels phospho-Zap70, and plotted as normalized to mean value of the 'control' cells. In the graph shown here, n1 = 34, n2 = 38, p = 0.16. (**C**) Synaptic phospho-PLCγ1 levels are reduced in cells lacking F-actin foci. DMSO (control, top panel) or CK666 (bottom panel) treated AND CD4 T cell blasts were incubated with bilayer containing anti-CD3 and ICAM1 for 2 min, fixed and stained with Alexa488-phalloidin (green) and phospho-Y783-PLCγ1 (red), and visualized using TIRF microscopy. In these images, total synaptic levels of phospho-PLCγ1 were assessed (top left, graph). These images were also analyzed using integrated morphometry and total number of phospho-PLCγ1 events per cell (c, top right, graph) or intensity distribution of the events across the population of T cells (bottom right, histogram) were measured. Note that CK666 treatment leads to a uniform reduction in phospho-PLCγ1 events across different intensity ranges (c, bottom histogram). In all graphs in c, n1 = 50, n2 = 67, p < 0.0001 (**D**) Total cellular levels of phospho-PLCγ1 and PLCγ1 with or without treatment of cells with CK666. CD4 T cell blasts were treated with 100 µM CK666 or DMSO (control), and then incubated with anti-CD3/CD28 beads for 5 min in the presence of the inhibitor, lysed and analyzed using western blotting. (**E**) In CK666-treated cells, there is a substantial reduction in synaptic phospho-PLCγ1 co-localized with F-actin foci. Colocalization analysis was performed on images from (**C**), to estimate phospho-PLCγ1 and F-actin colocalization, as described in 'Materials and methods'. (**F**) Pan- PLCγ1 levels at the synapse are reduced in CK666-treated cells. T cells were processed for PLCγ1 immunofluorescence and imaged as described above. In the graph, each dot represents PLCγ1 intensity per cell in the TIRF images. n1 = 29, n2 = 34, p < 0.0001. (**G**) Defective calcium ion mobilization in CK666 treated cells. AND CD4 T cell blasts were loaded with Fluo4, and treated with DMSO or 100 µM CK666 for 1 min, and then imaged live on anti-CD3 and ICAM1, to monitor calcium ion flux ('Materials and methods'). Each point on the graph (far right) represents mean value of the baseline corrected fluorescence ±SEM (n = 30 cells). (**H**) Nuclear translocation of NFAT1 in cells treated with CK666. DMSO (control, top panels) or CK666-treated (bottom panels) AND CD4 T cell blasts were incubated with glass coverslips coated with ICAM1 alone (left), or ICAM1 and anti-CD3 (right) for 10 min at 37°C, fixed and immunostained for NFAT1 (red), and stained with Alexa488-phalloidin (green). Cells were subsequently imaged using confocal microscopy. The middle sections from the z-stack of the images of the cells were used to quantitate nuclear levels of NFAT1, by outlining phalloidin-free central area (nucleus) of the cell. The graph (right) shows nuclear NFAT intensity in the nuclear area, in a single section per cell. n1 = 98, n2 = 40, n3 = 156, n4 = 31. p-values, p1 = 0.92, p2, p3 < 0.0001. (**I**) PLCγ1 dynamics at TCR microdots. Human CD4 T cells transiently expressing PLCγ1-YFP (PLC, green) were incubated with glass coverslips patterned with Alexa647 tagged anti-CD3 microdots (TCR, red) and coated with ICAM1 and imaged live using TIRF microscopy. The graphs on the bottom show linescan intensity profiles of PLCγ1 and anti-CD3 (TCR) across the pixels marked in the corresponding image in the top panels. The middle panel shows the intensity profiles during the addition of CK666, and the right panel shows the profiles 30 s after the addition of CK666. The graph in (**J**) shows anti-CD3 (TCR) and PLCγ1-YFP profiles obtained from at least 30 different microdots from >12 cells in each control and CK666 treatment background. PLCγ1-YFP expressing human T cells were incubated with patterned substrate for (1 + 4) min in the presence of DMSO (Control) or CK666, fixed and imaged using TIRF microscope. The CK666 was added after 1 min of incubation of cells with the substrate. The graph shows mean value of pixel intensities calculated from the linescan profiles. The pixel intensities within a linescan were normalized to lowest pixel intensity within that linescan prior to calculating mean value across different linescans. For a given treatment background and linescan profile, TCR and PLCγ1 intensities were obtained from the identical pixel positions. Scale bar, 5 µm.

The following figure supplements are available for figure 6:

**Figure supplement 1**. CK666 treatment leads to loss of foci-associated PLCγ1 and phospho-HS1 from the synapse.

**Figure supplement 2**. Effect of CK666 on TCR-induced Zap70 and PLCγ phosphorylation in human T cells.

**Figure supplement 3**. Arp2/3 complex inhibition and synaptic dynamics of Zap70 and PLCγ1 in live cells.

(*Brossard et al., 2005*; *Biggs et al., 2011*; *Roybal et al., 2013*). Our study thus defines a common F-actin rich module that bridges planar models and more complex, but more physiological, T-APC systems.

## F-actin foci across different T cell systems

F-actin 'foci' have been noted in prior studies (*Barda-Saad et al., 2005*; *Beemiller et al., 2012*), but they have not been studied in any detail. Ours is the first study, to our knowledge, to have performed a detailed characterization of foci as structures that are polymerized de novo via a WASP dependent mechanism. Previously, it was proposed that similar TCR associated F-actin foci were formed through a buildup of retrogradely flowing F-actin moving over artificially immobilized TCR on chrome micro-barriers in both Jurkat and primary T cells (*Smoligovets et al., 2012*; *Yu et al., 2010*). Further characterization of this buildup revealed that it is due to accumulation of pre-existing filaments (*Smoligovets et al., 2013*). We avoided this pitfall here by utilizing a system in which TCR MC remain mobile or are pre-established in micron scale spots that are segregated and are less likely to be traversed by F-actin flows, preventing an accumulation of filaments.

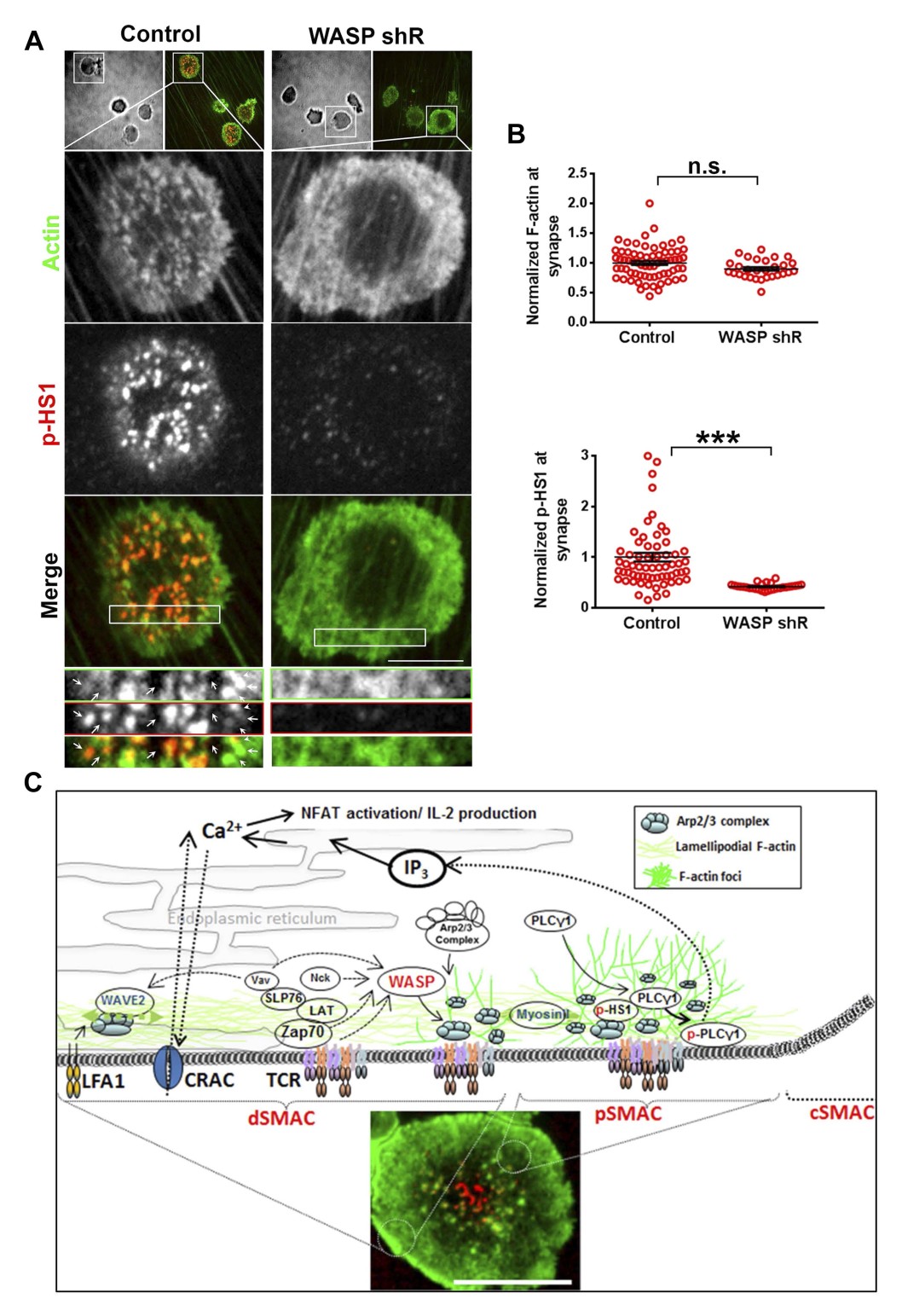

**Figure 7**. ILP signaling in T-EC immunological synapse is WASP dependent. Human CD4 T cells were incubated with culture media containing lentiviral particles carrying WASP shRNA or non-specific (control) shRNA for 48 hr (**A**) T cells transduced with WASP shRNA or control shRNA carrying lentiviral particles were incubated with endothelial monolayer for 10 min, fixed and processed for Alexa594-phalloidin (pseudo-colored green) and phospho-HS1 (pseudo-colored red) immuno-staining. The conjugates were then imaged using an EMCCD-coupled spinning disc confocal microscope. Each image represents a single confocal plane of T cell synapse, where the planar endothelial
*Figure 7. continued on next page*

*Figure 7. Continued*

interface is in focus. The area outlined in 'merge' panels was further scaled and magnified to show the details with more clarity (bottom panels). The top panels show the image of the field of view in DIC (left image) or fluorescence settings. (**B**) A reduction in WASP levels results in defective phospho-HS1 accumulation at T cell-endothelial cell synapse. The upper graph shows quantitation of phalloidin intensity in the synaptic plane, while the lower graph shows phospho-HS1 levels in the same plane. For both the upper and lower graphs, n1 = 68, n2 = 29, p1 = 0.071, p2 < 0.0001. This experiment was repeated twice with similar results. (**C**) Model of temporal sequence of events leading to F-actin foci formation and PLCγ signaling at the immunological synapse. Multiple pathways can result in actin polymerization and remodeling at the synaptic interface, contributing to F-actin organization in different SMAC zones. One such pathway involves WAVE2 recruitment by activated LFA1, followed by WAVE2 dependent Arp2/3 complex activation resulting in thick lamellipodial (dSMAC) and lamellar (pSMAC) F-actin meshworks. WAVE2-dependent F-actin pool is required for calcium-dependent calcium entry via the CRAC channel. Additional pathways including MyosinII-mediated actin remodeling is required for maintaining lamellar actin flow and directional persistence of microclusters (MCs) towards the cSMAC, and formin-mediated nucleation of F-actin promotes MTOC docking and stability of synapse. Another pool of F-actin or 'F-actin foci' is generated by the activity of WASP protein in the p- and dSMAC zones. Following TCR triggering, WASP is recruited at TCR signalosome via several possible mechanisms – such as via Vav, via NCK, via Zap70 and CrkL mediated WIP release and other effector mechanisms, and, through Fyn or PIP2 or PTP-PEST-binding at the plasma membrane (PM). Once activated, WASP recruits Arp2/3 complex to the MC, which then leads to actin branch nucleation and polymerization at the MC, over and above the local background actin. This process continues even during MC movement in the lamellar region, with a high F-actin turnover at the foci until its delivery to the cSMAC. In the foci, HS1 is recruited via binding both the Arp2/3 complex as well as F-actin, and is subsequently phosphorylated. As a consequence of early TCR signaling, PLCγ1 is also recruited to the MC signalosome, where it is stabilized via interactions with both F-actin, and foci residing HS1. F-actin foci dynamics in the proximity of the plasma membrane further support PLCγ1 phosphorylation, potentially by facilitating its interaction with PM-bound, upstream activators such as Itk. Phosphorylation of PLCγ1 by Itk then triggers phosphoinositide signaling, which in turn initiates calcium ion flux and NFAT1 activation. WASP deficiency or failure to activate Arp2/3 complex by WASPΔC mutant leads to selective loss of nucleation of foci at the MC. As a result, early signaling is not affected, however, both HS1 and PLCγ1 levels are severely reduced at the microcluster sites. The remaining PLCγ1 at synapse allows cell spreading and synapse formation, however, it is not sufficient to achieve calcium flux comparable to the control cells. Direct pharmacological inhibition of Arp2/3 complex using CK666 yields similar results; early TCR signaling is preserved while PLCγ1 phosphorylation and late signaling are severely perturbed. As actin polymerizing processes other than WASP also utilize Arp2/3 Complex, CK666-treated cells show a general reduction in lamellipodial and lamellar actin as well. However, the remaining F-actin levels are sufficient to support early TCR signaling. In contrast, total F-actin depolymerization at the synapse using CytochalasinD results in defects in early as well as late signaling, as has been reported in earlier studies. The image on the bottom shows a maximum intensity projection of synaptic contact interface of a human primary CD4 T cell, acquired using spinning disc confocal microscope. This cell was activated on a bilayer reconstituted with Alexa568 tagged anti-CD3 (red) and ICAM1 (unlabeled), for 2 min, fixed and stained for F-actin (green), and imaged.

The following figure supplement is available for figure 7:

**Figure supplement 1**. ILP F-actin and signaling in T-EC immunological synapse is dependent on Arp2/3 activity.

T cell ILP display molecular similarities to 'podosomal' actin structures that are a common feature of leukocyte contact with substrates (*Carman et al., 2007*; *Dovas and Cox, 2011*). Composed of dynamic F-actin, such podosomes are rich in HS1 related cortactin and are formed in a WASP-dependent manner at the adhesion interface of leukocytes (*Dovas and Cox, 2011*). We speculate that the TCR utilizes similar WASP-HS1 based molecular machinery to generate F-actin foci/ILP that then serve to support calcium ion signaling through stabilization of PLCγ recruitment and phosphorylation. In the immunoreceptor driven synapses of non-T cells, this foci-like organization of actin microfilaments has not been observed thus far. F-actin architecture at the NK cell synapse appears to be uniform and the deep lamellipodial network dominates the synaptic interface (*Rak et al., 2011*). Similarly, in B cells, a thick cortical actin meshwork that is inhibitory for early B cell receptor signaling has been reported (*Treanor et al., 2011*), but evidence of any 'foci-like' organization is missing. In such cell types, WASP could serve other roles, and may not promote immunoreceptor signaling via foci generation at the synapse.

In the case of both CD4 and CD8 primary T cells, foci are robustly triggered following TCR ligation. In activated mouse T cell blasts, antigen-dependent induction of foci was observed as well; however, a few

low intensity actin features were visible in a fraction of T cell blasts activated on ICAM1 in SLB (*Figure 4A*). We do not understand the origin of these low-intensity foci. It is possible that foci are generated at a basal level throughout the active signaling state of T cell blasts after initial priming, and antigen re-stimulation is a potent trigger for their formation. Alternatively, it is possible that integrins such as LFA-1 are capable of forming a few transient actin-rich features in activated lymphocytes in a specific chemokine environment, when its ligand ICAM1 is highly expressed. In contrast to the primary T cell systems, we have not detected foci in the Jurkat T cell line activated using the SLB system. A signaling network linking TCR to WASP has been outlined in the Jurkat cells (*Barda-Saad et al., 2005*; *Thrasher and Burns, 2010*), so it is likely this signaling pathway is intact. However, any WASP-dependent foci that may exist in these cells are either smaller than the resolution limit, or are transient in nature and rapidly dissipate into the lamellar network before they can be observed. Interestingly, in Jurkat T cells, the actin polymerizing function of WASP is not required for its role in calcium ion signaling (*Silvin et al., 2001*), and Arp2/3 silencing does not prevent PLCγ1 activation (*Gomez et al., 2007*). Thus, it is also possible that these cells utilize an alternative WASP-independent cytoskeletal pathway to support TCR-induced calcium ion elevation (*Babich et al., 2012*), and therefore do not need to utilize foci for this purpose. For example, the lack of negative regulators like PTEN may allow Jurkat cells to sustain calcium ion signals through alterations in lipid metabolism, without F-actin foci. A detailed comparison of the differential organization and function of F-actin modalities between Jurkat cells and the primary T cells will provide fundamental insights into the operation of compensatory cytoskeletal pathways that assist TCR distal signaling and may be important at different developmental stages in the T cell lineage or part of the failure of growth control in malignancy.

## Functional class of F-actin at the synapse

Our study revealed at least two functional classes of F-actin at the synapse. The first component is critical for very early signaling events encompassing TCR triggering and the proximal signaling cascade. This fraction is represented by a slow turnover CK666-resistant cortical F-actin pool, which most likely relies on filament remodeling proteins other than actin branching for its dynamics. The second component, the WASP and Arp2/3-dependent F-actin foci network is responsible for assisting PLCγ1 activation and calcium ion signaling. Furthermore, in addition to these two networks, there is evidence for a third, CK666-sensitive network that we have not investigated in this study. This network is made up of a fraction of lamellar and lamellipodial F-actin that is rapidly eliminated under even short duration treatments of CK666 based on the reduction in 'total F-actin' (*Figure 6—figure supplement 1B*). This observation indicates that this third sub-population is composed of dynamic F-actin. This network is not required for the early TCR signaling steps that we have tested, indicating that it may play other roles at the synapse that are known to be dependent on F-actin. For example, it may be equivalent to the WAVE2 dependent network that is not required for early TCR signaling or PLCγ1-dependent calcium ion efflux from endoplasmic reticulum, but is necessary for CRAC channel mediated calcium ion entry from extracellular medium (*Nolz et al., 2006*).

Considering that all the above-mentioned networks are made up of the same building blocks of G-actin, how are functional boundaries between them set up and regulated, and how is this related to the organizational differences between these networks? Given the small size of T cells, the highly dynamic nature of their F-actin structures and the pleotropic action of pharmacological inhibitors on all F-actin organizations, it is challenging to dissect the specific roles of these sub-populations during TCR signaling. Thus, further interrogation of these networks will require a combination of genetic tools enabling conditional perturbation, as well as rescue, of Arp2/3 complex and NPFs (*Moulding et al., 2007*; *Blundell et al., 2009*; *Sato et al., 2013*), in combination with super-resolution microscopy in primary T cells. The results from these experiments will reveal how these networks combine to generate an emergent multifunctional synaptic F-actin network during T cell activation.

## Is PLCγ1 the convergence point for multiple cytoskeletal pathways?

The primary recruitment of PLCγ1 from cytosol to the MC has been well characterized, and it involves its interaction with TCR signalosome proteins LAT and SLP76, and subsequent phosphorylation at PLCγ1 residue Y783 by membrane-bound Itk (*Park et al., 1991*; *Braiman et al., 2006*). Loss of F-actin foci resulted in reduced phospho-Y783 PLCγ1 levels at the synapse (*Figure 6C–D*). The microdot enrichment assays and the SLB experiments both indicated that this was largely due to defects in PLCγ1 accumulation at the MCs (*Figure 6*). Accumulation of PLCγ1 at actin foci could occur via its

direct association with actin filaments, as reported earlier (*Carrizosa et al., 2009*; *Patsoukis et al., 2009*), as well as with foci-recruited HS1. Both these mechanisms could simultaneously contribute to accumulation of PLCγ1 at TCR MCs.

The precise manner by which accumulation at actin foci promotes PLCγ1 activation is not clear. There are several possible mechanisms that could underlie this process. First, it is probable that the continual nucleation of F-actin filaments at foci provides a sustained cytoskeletal microenvironment for the enrichment of PLCγ1 around the TCR signalosome. As the TCR MC traverses regions with highly variable levels of lamellar F-actin en route to the cSMAC, the foci would provide for the continuous presence of high levels of F-actin at the TCR microcluster, which may in turn ensure continuous local enrichment of PLCγ1 at the MC. Second, foci F-actin growing in juxtaposition to the plasma membrane may facilitate the interaction of F-actin bound PLCγ1 with its plasma-membrane anchored effectors such as Itk, thereby enhancing its phosphorylation. In this model, the growth of foci would exert force on and push the plasma membrane in its vicinity. Consistent with this mechanism, we find that foci zones mark areas of high membrane tension, observed as the foci-dependent localization and increased phosphorylation of mechanosensory protein CasL (*Kumari et al., 2012*;; *Yu et al., 2012*) at the MC-foci sites (Santos et al., manuscript under review). This is also consistent with the manifestation as ILPs on the softer endothelial cell substrates. This observation also implies a possible role of F-actin foci in supporting synapse stability. Previously we have shown that WASP−/− T cell synapses form normally with normal pSMAC and dSMAC actin organizations, however they lost stability faster than those of WT cells (*Sims et al., 2007*). It will therefore be interesting to investigate whether the synapse stabilizing role of WASP is mediated by actin foci, and if foci influence pSMAC oscillations, as reported earlier in WASP−/− cells (*Sims et al., 2007*).

Interestingly, in CK666-treated cells, there is a modest initial rise in intracellular calcium ion concentration, but final calcium ion concentrations are significantly lower than in DMSO-treated control cells and rapidly decline thereafter (*Figure 6G*). The factors contributing to residual calcium ion elevation may be either residual phospho-PLCγ1 in CK666-treated cells, or an additional mechanism of PLCγ1 activation that is insensitive to this inhibition. In accordance with this, we do not observe a complete loss of PLCγ1 at microdot sites in CK666 treated cells. In contrast to the above mentioned processes that augment calcium ion signaling in mature T cells, there are alternative actin cytoskeletal pathways in immature thymocytes that impinge upon PLCγ1, and total F-actin is inhibitory for PLCγ1 activation and calcium ion flux (*Tan et al., 2014*). Detailed examination of F-actin networks in T cells during different developmental stages will clarify how the actin cytoskeleton can generate different activation outcomes at the level of PLCγ1 (*Kumari et al., 2014*; *Roybal et al., 2013*; *Malissen et al., 2014*).

## Unique requirement of WASP for foci generation

We find a specific requirement for WASP in the generation of F-actin foci. WASP−/− T cells show normal levels of total F-actin at the synapse, but no TCR MC associated F-actin foci. This lack of effect on total F-actin in WASP deficient or depleted cells may be attributed to the fact that the foci contribute to <10% of the total synaptic F-actin intensity (*Figure 1—figure supplement 1B*), and perhaps to a compensatory increase in the background actin networks when more ATP-G-actin is available for polymerization regulated by other NPFs such as WAVE2. We have shown that WASP's role in generating F-actin foci cannot be substituted for by NWASP. This selective requirement for WASP could be due to its specific interactions with upstream activators at the TCR signalosome. For example, a conventional mode of WASP and NWASP activation is through interaction with Cdc42 GTPase, However, WASP recruitment can occur independently of Cdc42 at the T cell synapse (*Cannon et al., 2001*). In agreement with this, in preliminary experiments, Cdc42−/− T cells showed normal F-actin foci (data not shown) indicating that these foci form without the aid of Cdc42. Additionally, Cdc42-dependent phosphorylation of WASP at Y291 by SFK is known to increase its affinity for Arp2/3 complex (*Torres and Rosen, 2003*). Overexpression of a phospho-deficient mutant (WASP Y291F) (*Blundell et al., 2009*; *Dovas et al., 2009*) did not perturb F-actin foci formation or PLCγ1 phosphorylation, indicating that the WASP activation pathway utilized for foci formation may be distinct from the upstream regulatory pathways involving Cdc42 activation, or SFK binding at phospho-Y291. While it was not possible to dominantly uncouple TCR MC from PLCγ1 activation by over-expressing WASP that is able to activate Arp2/3, but does not interact with SFKs, it is possible that other regulatory phosphorylation sites can scaffold WASP's interaction partners, thereby playing an ancillary role in PLCγ1 activation (*Torres and Rosen, 2003*; *Blundell et al., 2009*;

*Macpherson et al., 2012*; *Reicher et al., 2012*). These results, and the failure of compensation by NWASP, suggest that WASP is activated and employed in a specialized manner in T cells to generate F-actin foci. The dimerization property of WASP and positive feedback with Arp2/3 complex (*Padrick and Rosen, 2010*), as well as its role in polarizing micron-scale 'lipid rafts' at the synapse (*Dupre et al., 2002*) may contribute towards a specific enrichment and involvement of WASP at foci. At this stage, the mechanistic basis for WASP vs NWASP utilization is not understood.

To summarize, we have characterized a subsynaptic F-actin organization in T cells that is functionally distinct from larger scale lamellipodial and lamellar networks. These actin foci provide a signaling scaffold that is required for TCR-distal signaling and play a critical regulatory role in calcium ion signaling. There are significant questions that remain to be answered. For example, we do not yet know the MC-intrinsic properties that temporally regulate F-actin foci formation. We find that ~40% of TCR MCs are associated with F-actin foci at steady-state. While a large fraction of remaining clusters are in the central cSMAC region of the cell, previously characterized to be an actin depleted zone containing signaling incompetent TCR (*Kaizuka et al., 2007*; *Vardhana et al., 2010*), a smaller fraction of foci-unassociated MC resides in the lamellar zone as well. At lamellar zone TCR MCs, it is unclear which MC-intrinsic attributes dictate foci induction, maintenance and subsequent termination of actin nucleation on entry into the cSMAC, and if integrin signaling assists in any of these steps. Further investigation will not only provide a deeper understanding of subsynaptic F-actin dynamics, but will also illuminate other specialized roles that F-actin subpopulations play during the lifetime of a signalosome. The results will also provide deeper insights into signaling compartmentalization of the T cell synapse that is regulated by the actin cytoskeleton and its alteration in WAS pathology.

## Materials and methods

### Cell culture and transfection

AND Mouse T cells used in this study were isolated from 2–3 months old AND B10.Br mouse spleen and lymph nodes. For experiments involving activation with bilayer containing MHC-peptide, AND CD4 T cell blasts were used. Briefly, total splenocytes and lymph node cells were expanded in vitro, in the presence of 2 µM MCC peptide and 10 units/ml IL-2 for 6 days in Kanagawa's modified Dulbecco's modified Eagle Medium (KDMEM) supplemented with FBS, and were subsequently used for experiments. In other experiments freshly isolated CD4 T cells were used, as indicated in figure legends. For experiments with mouse CD4 T cells, spleens and lymph nodes were harvested, and CD4 T cells were purified using mouse CD4 isolation kit (Miltenyi Biotec, Auburn, CA). These CD4 T cells were used immediately after isolation for the experiments. For WASP single knockout experiments, *Was−/−* C57BL/6J mice (stock number 019458) were purchased from Jackson Laboratory (Bar Harbor, Maine), and age-matched WT C57BL/6J mice were also purchased as controls (Stock number 000664). For WASP and HS1 single knockout, and *Was−/− Hcls1−/−* double knockout studies, C57BL/6J, *Hcls1−/−*, *Was−/−* and *Hcls1−/− Was−/−* mice (*Dehring et al., 2011*) were used, and CD4 T cells purified as described above. For experiments with WASP−/−, NWASP−/− and WASP−/− N-WASP−/− CD4 T cells, cells were purified from spleen and lymph nodes of WT 129 or knockout 129 mice (obtained from Snapper laboratory [*Cotta-de-Almeida et al., 2007*]), as described above. For experiments with human CD4 T cells, cells were purified from fresh peripheral blood samples (New York Blood Center, NY) using Rosette Sep kit (Stemcell Technologies, Vancouver, BC, Canada) according to the manufacturer's protocol. Human CD4 T cells were rested overnight in RPMI supplemented with FBS, prior to their use in experiments.

### Reagents

N-WASP (H-100) and WASP (H-250) antibodies were purchased from Santa Cruz Biotechnology (Dallas, TX). HS1 antibody (D5A9), Phospho-Y397 HS-1 antibody (D12C1), phospho-Y319 Zap70/Y352 Syk (affinity purified antisera #2704), PLCγ1 (D9H10), phospho-Y783 PLCγ1 (#2821), NFAT1 (D43B1), phospho-Y416 SFK (#6943), phospho-Y171 LAT (#3581) and WAVE2 (D2C8) antibodies were obtained from Cell Signaling Technology (Beverly, MA). Anti-Arp3 monoclonal antibody (FMS338) and anti-MyH9 (Myosin IIA, #SAB2101542) antibody were purchased from Sigma Chemicals (St. Louis, MO). Rabbit p34-Arc (anti-Arpc2) antibody was purchased from EMD Millipore and phospho-Y145 SLP76 (#NBP140511) was purchased from Novus Biologicals. Celltracker-CM-DiI was purchased from Life Technologies. Anti-rabbit secondary antibodies conjugated with Dylight647 and Dylight564 dyes or anti-mouse Fc-specific Alexa568 secondary antibodies were procured from Jackson Immunoresearch (West grove, PA) and

tested for lack of cross-reactivity prior to their use in assays. For anti-TCR antibodies-mediated T cell activation on bilayers, 145-2C11 (#16-0031, for mouse cells) and OKT3 (#16-0037 for human cells) clones were purchased (eBioscience, San Diego, CA), mono-biotinylated and labeled with fluorophores, and tested for mono-biotinylation prior to use in experiments. For experiments involving assessment of PLCγ1 activation using western blots, T cells were activated with anti-CD3/CD28 coated Dynabeads (Life Technology, Grand Island, NY, #11452D) according to manufacturer's instructions. A plasmid encoding Lifeact-GFP was a kind gift of B Geiger (Weizmann Institute, Rehovot, Israel), and GFP-tagged WASP constructs were a kind gift of Dianne Cox (Albert Einstein School of Medicine, New York). Blebbistatin, Wiskostatin and PP2 were obtained from Sigma chemicals; CK666 and CK689 were obtained from Calbiochem (EMD Millipore, Billerica, MA). Alexa488-phalloidin and Alexa568-phalloidin were purchased from Invitrogen (Life Technologies). Other fluorescent probes were obtained from Invitrogen and were conjugated with desired proteins according to the manufacturer's protocols. I-E$^k$-6His and mouse ICAM-1-12His were produced in the S2 insect cell line. I-E$^k$-6His was either loaded with moth cytochrome C peptide 91–102 or in some cases was the same MCC peptide was covalently coupled to the β chain. The density I-E$^k$-6His and ICAM1-12His was 20–50 molecules/μm$^2$ and 200 molecules/μm$^2$, as described previously (*Grakoui et al., 1999*; *Choudhuri et al., 2014*), unless otherwise noted.

## WASP silencing and western blotting

For siRNA experiments, siRNA were obtained from Dharmacon (Thermo Fisher Life Sciences, Pittsburgh PA). Mouse WASP siRNA (on-target plus, Cat no-LU-046528-01-0005), Mouse HCSL-1 (HS1, on-target plus, Cat no-LU046134-01-0002) were purchased as a set of four. A mixture of these four siRNAs was used at a concentration of ~600 nM for 5 million cells, for WASP, HS1 and non-specific (control) siRNA (D-001810-10-20) cases. Cells were electroporated using Amaxa Nucleofector (Lonza, Basel, Switzerland) following the instructions from the manufacturer, and grown in K-DMEM for 72–80 hr before experiments. For lentiviral transduction experiments, GIPZ lentiviral expression system (Thermo Scientific, Pittsburgh PA) was used. Briefly, the HEK cells were transfected with GIPZ-WASP shRNA plasmids (RHS4287-EG7454, a mixture if three sequences-5′ TTCTTATCAGCTGGGC TAG 3′ (V2LHS_92677), 5′ TTGCTGATCTTCTTCTTCC 3′ (V2LHS_92678), and 5′ TCATCTTCATCGC CAGCCT 3′ (V3LHS_352271)), or non-specific shRNA plasmid (RHS4346) along with the trans-lentiviral packaging mix (RHS4287-EG7454), using the calcium phosphate method. The culture supernatant containing viral particles was isolated after 68 hr of transfection, and utilized to transduce CD4 T cells according to manufacturer's instructions. The transfected T cells were identified by GFP expression in both control and WASP shRNA transduction cases.

WASP depletion was assessed using western blots. Approximately 2 million cells from the control and siRNA samples were harvested, lysed using RIPA buffer supplemented with protease inhibitor, and the resulting lysates were resolved in reducing conditions on a polyacrylamide gradient gel (Biorad, Hercules, CA). After transfer to the nitrocellulose membrane, protein levels were assessed by staining with primary (overnight at 4°C) and IR-dye conjugated secondary antibodies, and subsequently imaged using an Odyssey infrared imager (LI-COR Biosciences Lincoln, NB).

## Bilayer experiments and immunostaining

CD4 T cells were incubated with bilayer reconstituted with MHCp (I-E$^k$-MCC) and ICAM1, or 2C11 and ICAM1 (AND mouse CD4 T cells and polyclonal C57BL/6J T cells), or with bilayer reconstituted with OKT3 and ICAM1 (human polyclonal CD4 T cells) for the indicated time duration in Flow chambers (Bioptech Inc., Butler, PA) or Ibidi microscopy chambers (Ibidi, Verona, WI). Bilayer deposition and ligand density determination was performed as described previously (*Grakoui et al., 1999*; *Ilani et al., 2009*; *Choudhuri et al., 2014*). For immunostaining, cells were fixed using PHEMO buffer (10 mM EGTA, 2 mM MgCl$_2$, 60 mM Pipes, 25 mM HEPES, pH 6.9) supplemented with 3.7% para-formaldehyde at room temperature for 20 min and permeabilized using 0.1% Triton X-100 for 2 min. Cells were then blocked using 5% casein for 30 min, washed and incubated with primary antibody overnight at 4°C and appropriate secondary antibodies for 1 hr, followed by washing and visualization using the indicated microscopy method.

For all of the colocalization experiments, mouse AND T cell blasts activated on MHCp bilayers were utilized. For WASP knockout T cell experiments, freshly isolated T cells were utilized. For all live imaging experiments, peripheral human CD4 T cells were utilized, unless otherwise noted.

## Imaging with calcium ion sensitive dye

AND CD4 T cell blasts were loaded for 30 min with 2 µM of Fluo4 (Life Technologies) in HEPES buffered saline (HBS) which was supplemented with 1% HAS and pluronic acid. Cells were then treated with DMSO or 10 µM CK666 for 1 min, and then incubated in glass chambers (Nunc Lab-Tek, Thermo Scientific, Rochester, NY) containing 2C11 and ICAM1. Cells were imaged for the indicated time, using a confocal microscope with fully open pinhole. The Fluo4 fluorescence obtained for each cell was normalized to baseline fluorescence, and mean values from individual cells were then plotted on the graph ±standard error of the mean (SEM).

## Anti-CD3 (OKT3) microdot substrates

Patterns of OKT3 antibody were created by micro-contact printing onto glass coverslips. First, topological masters were fabricated by electron beam using a 1 µm PMMA layer, and spun-coated onto silicon wafers. Then, a combination of a thin layer of hard PDMS and a thick layer of Sylgard184 was spun-coated onto these wafers, resulting in hydrophobic stamps for patterning. Stamps were incubated with 1:1 ratio of Alexa647-OKT3 and unlabeled isotype antibodies, maintaining a total concentration of 25 µg/ml. Following coating, stamps were rinsed with PBS, PBS + 0.05% Tween-20, and deionized (DI) water, dried with $N_2$ gas, and then placed in contact with plasma-cleaned coverslips for 2 min. Coverslips were washed and incubated with 1 µg/ml ICAM1 for 40 min, after which they were washed again and then used for the experiment.

## Barbed end labeling during actin polymerization

Barbed end labeling of F-actin was carried out as previously described (*Furman et al., 2007*). Briefly, T cells were incubated with glass coverslip coated with ICAM1 and anti-CD3 for 5 min, and then incubated with 0.0125% saponin dissolved in Rinse buffer (20 mM HEPES, 138 mM KCl, 4 mM $MgCl_2$, 3 mM EGTA, pH 7.5) along with freshly dialyzed Rhodamine labeled G-actin (0.5 µM in chilled G-actin buffer-4 mM TRIS, 0.2 mM CaCl2, 0.2 mM ATP, 1.0 mM DTT) for 1 min at 37˚C. They were then rapidly fixed using 0.5% glutaraldehyde dissolved in cytoskeletal buffer (10 mM MES, 3 mM MgCl2, 138 mM KCl, 2 mM EGTA) for 10 min. Following this, cells were washed and permeabilized using 0.5% X-100 dissolved in cytoskeletal buffer for 5 min, and then incubated with sodium borohydride (100 mM made fresh in PBS) for 10 min. Subsequently, cells were processed for total F-actin staining, using Alexa488-phalloidin, mounted using Fluoromount, and visualized using TIRF microscopy.

## Cell–cell conjugate experiments

As described previously (*Sage et al., 2012*), human dermal microvascular endothelial cells (HDMVECs) were cultured on fibronectin (10 mg/ml) in complete EBM-2 MV media (Lonza, MA). ECs were transfected with palmitoylated-YFP DNA construct ('membrane-YFP', from Life Technologies, CA) using Amaxa electroporation according to manufacturer's instructions. Prior to the experiments, ECs were activated for 48 hr with 100 ng/ml IFNγ and 12 hr with 50 ng/ml TNFα. ECs were eventually pulsed with 1 mg/ml SEB and TSST for 60 min (sAg, Toxin technologies, FL), incubated with human peripheral T cells (sAg expanded T cells, cultured in presence of IL-15) for the relevant time duration, and imaged using either an upright laser scanning confocal microscope (Zeiss) or spinning disc confocal microscopy (Andor technology CT).

## Microscopy and image analysis

For TIRF microscopy, imaging was carried out using a Nikon Eclipse Ti microscope with a 100× 1.49 NA objective, and an AndorDU897 back illuminated EMCCD camera. For spinning disc confocal microscopy, an Olympus IX70 microscope with a 150× 1.45 NA objective, CSU (Solamere Technology, Salt Lake City, UT) and a Hamamatsu Orca ER camera was used. Solid-state lasers (Coherent, CA) provided illumination at 488, 561 and 641 nm, and narrow pass filters (Chroma Technology, Burlington, VT) were used for detection.

All image processing was carried out using ImageJ and Metamorph software. For colocalization analysis, raw images were subjected to local background subtraction as follows. Raw images were subjected to a local 50% intensity rank (median) filter (1.6 × 1.6 microns box size, see *Figure 1—figure supplement 1*,) across the whole image in order to generate a 'background image'. This background image was then subtracted from the raw image. The resultant high-pass filtered image was thresholded,

and used to assess co-localization using similar methodology to assess pixel overlap across overlaid thresholded images as described before (*Kumari et al., 2008*), via 'co-localization class' plugins in ImageJ. The intensity of overlapping pixels for each cell were normalized to total intensity of the same cell in the thresholded images, and were plotted as 'fraction co-localized' per cell (*Figure 2—figure supplement 1*).

To assess F-actin foci alone, raw images were subjected to 50% rank filtering (1.6 × 1.6 microns box size, subsample ratio 1) and resultant image was subtracted from original image (*Figure 1—figure supplement 1*). All the samples were subjected to identical filter settings prior to analysis. Cell intensities from the subtracted image were then calculated by marking regions around the cells, and using 'regional measurements' features of MetaMorph Software. In most graphs 'intensity' or 'levels' on Y-axis refers to integrated intensities in the cell synapse, unless otherwise indicated.

The images are displayed at identical contrast settings, and all the images presented here are raw images, unless otherwise mentioned. All the experiments were repeated at least thrice unless otherwise mentioned. For the graphs showing F-actin foci intensity, the datasets were normalized to the average value from 'control' data for that experiment, and plotted as 'normalized intensity' unless otherwise mentioned. For statistical analysis, the Mann–Whitney unpaired t-test was performed, unless otherwise mentioned. The bars and error bars in all graphs represent mean ± SEM.

## Acknowledgements

We thank A Babich, Fiona Clarke and J Goettel for invaluable help with the HS1 and WASP/NWASP knockout cells respectively. We thank S Snapper for generously providing WASP/NWASP knockout cells, L Samelson for PLCγ1-YFP plasmid construct, Dianne Cox for GFP-tagged WASP constructs, A Dhawale for critical reading of the manuscript, Russell McConnell (Gertler laboratory) and the Dustin laboratory for helpful discussions. We thank the Keck imaging facility at the Whitehead Institute. This work was supported in part by the NIH Common Fund through a Nanomedicine Development Center PN2EY016586 (MLD), and NIH NIAID through R37AI043542 (MLD), R01AI65644 (JKB), R01AI088377 (LCK), the NYULMC Office of Collaborative Science Microscopy Core, Cancer Research Institute (SK), and a grant from the National Psoriasis Foundation. DJI is a Howard Hughes Medical Institute investigator and MLD is a Principal Research Fellow of the Wellcome Trust and Senior Research Fellow of the Kennedy Trust for Rheumatology.

## Additional information

### Funding

| Funder | Grant reference number | Author |
|---|---|---|
| National Institute of Allergy and Infectious Diseases (NIAID) | R37AI043542 | Michael L Dustin |
| National Institute of Allergy and Infectious Diseases (NIAID) | R01AI65644 | Janis K Burkhardt |
| National Institute of Allergy and Infectious Diseases (NIAID) | R01AI088377 718 | Lance C Kam |
| National Institutes of Health (NIH) | NIH common fund-Nanomedicine Development Center (PN2EY016586) | Michael L Dustin |
| Cancer Research Institute | Postdoctoral fellowship | Sudha Kumari |
| National Psoriasis Foundation | Annual grant | Michael L Dustin, Sudha Kumari |
| New York University Langone Medical Center | NYUMC microscopy core | Michael L Dustin |

The funders had no role in study design, data collection and interpretation, or the decision to submit the work for publication.

## Author contributions

MLD, JKB, Conception and design, Analysis and interpretation of data, Drafting or revising the article, Contributed unpublished essential data or reagents; SK, Sudha Kumari designed the experiments, acquired and analyzed the data and wrote the manuscript., Conception and design, Acquisition of data, Analysis and interpretation of data, Drafting or revising the article; DD, Conception and design, Acquisition of data, Analysis and interpretation of data; RM, CVC, EC-T cell conjugate experiments, Conception and design, Acquisition of data, Analysis and interpretation of data; EJ, LCK, Conception and design, Contributed unpublished essential data or reagents; GC, Actin barbed end labeling experiments, Conception and design, Acquisition of data, Analysis and interpretation of data; FBG, Conception and design, Analysis and interpretation of data; DJI, Analysis and interpretation of data, Contributed unpublished essential data or reagents

## Author ORCIDs

Michael L Dustin, http://orcid.org/0000-0003-4983-6389

## Ethics

Animal experimentation: This study was performed in strict accordance with the NIH guidelines for the care and use of laboratory animals. The protocol was approved by the committee on animal care (CAC), CAC protocol # 0714-073-17, division of comparative medicine at the Massachusetts Institute of Technology.

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
