## [Decision Letter]

Thank you for sending your work entitled “WASP dependent actin foci facilitate phospholipase C-γ activation in primary T lymphocytes” for consideration at *eLife*. Your article has been favorably evaluated by Tadatsugu Taniguchi (Senior editor), a Reviewing editor, and two reviewers.

The Reviewing editor and the reviewers discussed their comments before we reached this decision, and the Reviewing editor has assembled the following comments to help you prepare a revised submission.

The referees found your paper of substantial interest to the readers of *eLife* and feel that with suitable revision, it would be an excellent contribution. The major issues raised by the reviewers that need to be dealt with in your revision are detailed below, along with a few minor comments. A revised paper addressing in a substantive manner the major issues raised will be examined by the reviewing editor and the referees and final decision made at that time.

S. Kumari et al. employ a combination of cutting-edge microscopy techniques to investigate the role of Wiscott Aldrich Syndrome protein (WASP) in T cell activation. The authors have developed a novel assay that reveals actin foci that form after activation of the T cell receptor. These have not been noticed before, since this newly polymerized actin represents just a subset of the total actin filament pool. The discovery of distinct types of actin filaments in the T cell is interesting in its own right. These actin foci are associated with TCR microclusters and require WASP and HS-1 for their formation. Interestingly, disruption of the formation of these foci by WASP depletion or Arp2/3 inhibition with CK666 results in the inhibition of PLCg activation and calcium mobilization; however, proximal signaling is not affected. These results provide new insight into the role of actin networks in signaling and also provide a potential explanation of the immunodeficiency defected associated with mutations or knockdown of WASP. This is a very comprehensive and well-conducted study (perhaps a bit dense read in places).

Major issues:

1) The authors used a variety of methods of stimulating T cells (pMHC and other times with two different antibodies (Okt3 or 2C11), and also switched between mouse and human T cells. Mouse and human T cells were even employed in the same figure (e.g., Figure 4). The explanation for this is not clear in the paper and would be good if it was explained to the reader. Also, in several parts of the paper, the authors are not internally consistent in how they conduct their experiments, without explanation for the change in method. For example, CD3 antibody was used to activate the T cells when studying the role of WASP in forming actin foci, but pMHC was used for investigating the role of NWASP and WAVE2. Why the two different types of stimuli in different experiments? Unless the authors have a strong experimental or theoretical basis for the switch, these experiments need to be made congruent.

2) The expression of WASP mutants was used to dissect the role of WASP (e.g. Y291F and deltaC). However, the results are somewhat complicated to interpret. Ideally, it would have been best to express these in WASP-/- T cells, which they employed in panel a-c of the same figure (Figure 4). This would rule out potential complications with having endogenous WASP also present (particularly important for the lack of an effect of Y291F expression). Also I do not understand how Y291F serves as a control. This phosphorylation should enhance WASP activation and increase affinity for Arp2/3. Thus, a mutant would be impaired in these activities but still would retain a scaffolding function. Thus, isn't it still possible that WASP activates PLCg by acting as a scaffold? Again, please make the experiments internally consistent (e.g., use WASP-/- cells throughout as recipient cells) and explain the choice of Y291F as a control.

3) In Figure 3, the authors determined whether TCR signaling is required for formation of the actin foci. They have compared “ICAM alone” and “ICAM + MHCp” conditions. It would be useful to include the “MHCp alone” condition as well to determine if integrin signaling is also involved.

4) The Arp2/3 inhibitor CK666 abolished 67% actin foci and 42% of lamellipodia and lamellar actin. While the foci are marginally more reduced, it is difficult to attribute the defects in calcium flux and NFAT translocation (Figure 6) to foci reduction rather than lamellipodia and lamellar actin reduction. It might be better to test these downstream signaling defects in WASP deficient cells because depleting WASP appears to be more selective for reducing foci and does not affect lamellar actin structures as much. Further, at least some of the results obtained using CK666 should be confirmed with a more selective approach based on Arp2/3 silencing.

5) The results showing the role of WASP on foci formation are obtained using fixed and permeabilized cells at a defined time point (2 minutes after beginning of culture). This makes it difficult to estimate the time of formation of these structures. It would be important to investigate the time kinetics of foci formation and of their migration towards cSMAC using time-lapse video microscopy to define whether the kinetics are altered (delayed) upon WASP/Arp2/3 silencing. It would also be interesting to monitor the kinetics of foci formation and of their lifetime in different areas of the synapse (independently of WASP silencing) in parallel with [Ca^2+^] i increases. This approach would allow to establish whether foci formation precedes or not the [Ca^2+^] i increase.

6) The authors propose that WASP pathway regulates actin foci formation, which in turn is instrumental for synaptic activation of PLCγ1. However, they do not show a mechanistic link between foci formation and PLCγ1. An alternative explanation of the data might be that the WASP pathway might control PLCγ1 activation. In turn PLCγ1 signaling might promote actin dynamics as suggested by a recent report (A. Brodovitch et al., J Immunol, 2013; 191:2064-2071). To address this point, the authors should investigate the impact of PLCγ1 silencing (or of its pharmacological inhibition) on F-actin foci formation and the time kinetics of foci formation and of [Ca^2+^]i (see above).

---

## [Author Response]

The reviewers have made excellent suggestions that we have addressed by carrying out new experiments, as well as providing better explanations for the rationale and extensive literature behind our existing experiments.

*1) The authors used a variety of methods of stimulating T cells (pMHC and other times with two different antibodies (Okt3 or 2C11), and also switched between mouse and human T cells. Mouse and human T cells were even employed in the same figure (e.g.,*
Figure 4*). The explanation for this is not clear in the paper and would be good if it was explained to the reader. Also, in several parts of the paper, the authors are not internally consistent in how they conduct their experiments, without explanation for the change in method. For example, CD3 antibody was used to activate the T cells when studying the role of WASP in forming actin foci, but pMHC was used for investigating the role of NWASP and WAVE2. Why the two different types of stimuli in different experiments? Unless the authors have a strong experimental or theoretical basis for the switch, these experiments need to be made congruent*.

We agree that the use of multiple experimental systems has the potential to become a distraction if the advantages of each system are not explained clearly. We found that the key features of the foci pathway i.e. the mechanism of foci formation via the WASP/HS-1 dependent pathway and its impact on PLCγ activation are conserved in human and mouse T cells, activated using a variety of activation stimuli (Figure 1—figure supplement 3, Figure 1—figure supplement 4 and Figure 1—figure supplement 5, Figures 2, 3, 4, 5, 6 and 7, Figures 8 and 9). This data is consistent with the previous reports, where TCR-dependent calcium signaling was found to be dysregulated in both WASP-/- mouse as well as WAS human T cells (Introduction). We thus utilized these conserved features of the pathway to overcome the technical limitations of individual experimental systems. As such, a large number of the presented experiments were replicated in both human and mouse antigen and agonist antibody-based systems. However, due to the methodological limitations in some contexts, as outlined below, some of our experiments were more suitable to perform in a specific system.

Author response image 1.CK666 treatment results in reduced phosphorylation of PLCã1 at the T cell synapse activated on MHCp. AND CD4 T cell blasts treated with DMSO (Control, upper panels) or CK666 (lower panels) were incubated with bilayers containing ICAM1 and MHCp, for 2 min at 37⁰C. Cells were then fixed, immunostained with anti-phospho-PLCã antibody and Alexa488-phalloidin, and visualized using TIRF microscopy. The graph shows phosphoPLCã levels at synapse, in control and CK666 treated cells. n1=47, n2=60, p <0.0001.**DOI:**
http://dx.doi.org/10.7554/eLife.04953.035

*Peptide-MHC vs. 2C11-based activation*:

Activation of TCR transgenic CD4 T cells using agoinst peptide-MHC is the most physiological mode of stimulation with SLB as both TCR and CD4 are engaged with natural ligands. This system has been routinely utilized for high resolution characterization of cytoskeletal organization at T cell immune synapse ([3]; Hashimoto-Tane et al., 2011; [44]). Using siRNA-mediated silencing, we found that the mechanism of foci formation was indeed WASP dependent in this system (30% reduction in foci upon partial WASP silencing, data not shown), and was comparable to foci induction in cells activated using anti-CD3 (Results section, paragraph 1, 27% reduction in foci in T cells activated using anti-CD3, *p=0.0025* (Figure 1—figure supplement 3); 30% reduction in foci in T cells activated using MHCp, *p=0.0005* (data removed now), upon partial WASP silencing]. We have now replaced the MHCp data with anti-CD3 data, for a more coherent presentation of results. We continued to employ MHCp-based activation for most colocalization experiments in the paper, as this system provided cleaner staining with some antibodies, including the mouse monoclonal anti-Arp3 antibody (Figure 5), which would otherwise cross-react with agonist 2C11 (hamster in origin) antibody, and NWASP and WAVE2 antibodies (Figure 9). In addition, since this system utilized activated CD4 T cells, which are larger and more suitable for transfection procedures than primary T cells, we were able to achieve higher transfection and survival rates for siRNA experiments, and better spatial resolution for colocalization experiments.

Author response image 2.Barbed-end labeling of polymerizing actin filaments in Human CD4 T cells. T cells were incubated with coverslips coated with anti-CD3 and ICAM1 for 5 min, and then processed for barbed end labeling and visualized using TIRF microscopy as described in Methods and Figure 1. The images show total F-actin (Alexa488-phalloidin, left image) and barbed ends (Rhodamine-phalloidin or ‘fresh F-actin’, middle image) labeled within 1 min of actin polymerization at the synapse. The foci areas in the total F-actin image were identified and outlined by intensity rank-based filtering. These outlined areas were then analyzed in both ‘fresh F-actin’ (barbed end labeled), and ‘total F-actin’ images. The average intensity per pixel within foci regions (‘F-actin foci’), and outside the foci area (lamellipodial surround), was measured and plotted (bottom right graph). The bottom left graph shows the normalization of fresh F-actin (graph on the right) with total F-actin in the same area. n=17 cells, for right graph, p<0.0001, for the graph on the left, p=0.9.**DOI:**
http://dx.doi.org/10.7554/eLife.04953.036

The disadvantage of the peptide- MHC approach is that it requires maintaining all knockout strains with the TCR transgene. The WASP, NWASP and HS1 deficient mice were not maintained with the TCR transgenes and were polyclonal 129 or C57B6 backgrounds (Cannon et al., 2004). Thus, we utilized the agonist anti -CD3 antibody (2C11) to activate the freshly isolated primary CD4 T cells from control and gene targeted mice and only made comparisons in the same strain backgrounds, for a number of experiments demanding complete removal of gene products. Although anti-CD3 based activation is a more synthetic way of triggering TCR, it has frequently been utilized interchangeably with the antigen-specific system for assessing the role of actin regulatory proteins in T cell activation (Bouma et al., 2011; Cannon et al., 2004; [32]; Moulding et al., 2013; [54]; Padrick et al., 2010; [68]; [69]). Indeed, we found that the mechanism of foci formation, and the effect of their depletion on PLCγ1 activation are comparable between the 2C11 and peptide -MHC systems (Figure 6, Figure 8), thus justifying the use of both systems for complementary experiments. In fact we have attempted to simplify the initial data presentation to focus on the results with mouse knockout stains with anti-CD3 stimulation only (Figure 1), later introducing other systems with appropriate stimulation (Figure 2).

*Human T cell activated on OKT3*:

For live imaging and overexpression studies, we could not continue to utilize mouse cells, and switched to human cells for two reasons. First, primary mouse T cells could not recover from electroporation utilized for transfecting them, as noted previously (17). Although activated mouse T cells would survive better than primary T cells after transfection procedures, we were reluctant to generate WASP-/- T cell blasts for overexpression experiment, given previously reported activation and proliferation defects in WASP-/- T cells. Second, human WASP constructs have already been utilized successfully for dominant negative inhibition experiments in human model T cell line (71). Therefore, in Figure 4, as pointed out by the reviewer, instead of primary mouse T cells, primary human CD4 T cells were utilized to assess the effects of human WASP dominant negative constructs. Independently, we ascertained the dynamics of foci at synapse (Figure 9), the effects of WASP depletion on foci formation (now presented in Figure 2—figure supplement 2), and their role in early and late TCR signaling in human T cells (now presented in Figure 6—figure supplement 2 and Figure 6—figure supplement 3, Figure 7, Figure 4), and observed outcomes identical to those in mouse cells (Figure 1, Figure 6, Figure 8).

We agree with the reviewers, that in the case of WAVE2 and NWASP recruitment, the colocalization data should have been performed in an identical system as the Figure 1, utilizing anti-CD3 for T cell activation. We did attempt to use anti-CD3 based primary mouse T cell based system previously, however the anti-WAVE2 antibody (H-110, Santacruz Biotechnology) (Goh et al., 2012), gave high background on the 2C11 coated surface (Figure 10). Thus, we had utilized peptide-MHC-based activation, which substantially reduced the non-specific background staining. Following the reviewer’s suggestion, we now utilized a different antibody (D2C8, Cell Signalling technology) (Wernimont et al ., 2011), which provides cleaner labelling of endogenous WAVE2 in the cells activated using anti-CD3 (Figure 10). We have now provided this new experimental data, to replace the old set (Figure 1–figure supplement figure 4B) . The current data shows a comparable lack of colocalization between actin foci and NWASP/WAVE2 (NWASP 9.48%, WAVE2 14.36%), as we reported using pMHC system previously (NWASP 7.54%, WAVE2 11.7%).

Author response image 3.A comparison of endogenous WAVE2 staining, obtained using two different rabbit anti-mouse WAVE2 antibodies, and identical immunostaining procedure as described in Methods. Primary mouse CD4 T cells were activated on anti-CD3 and ICAM1 coated surface for 5min, fixed and processed for indirect immunofluorescence using H-110 clone (upper panels) or D2C8 clone (lower panels) and Alexa647 tagged secondary donkey antirabbit antibody. Cells were also stained with Alexa488-phalloidin. Note that H110 staining results in high fluorescence background in the areas surrounding cells (arrows). This pattern of staining was observed in two independent experiments.**DOI:**
http://dx.doi.org/10.7554/eLife.04953.037

We have now provided the explanations for a switch of the systems in the main text at various places, as well as a brief explanation in the beginning of the Methods section, as suggested by the reviewers.

*2) The expression of WASP mutants was used to dissect the role of WASP (e.g. Y291F and deltaC). However, the results are somewhat complicated to interpret. Ideally, it would have been best to express these in WASP-/- T cells, which they employed in panel a-c of the same figure (*Figure 4*). This would rule out potential complications with having endogenous WASP also present (particularly important for the lack of an effect of Y291F expression). Also I do not understand how Y291F serves as a control. This phosphorylation should enhance WASP activation and increase affinity for Arp2/3. Thus, a mutant would be impaired in these activities but still would retain a scaffolding function. Thus, isn't it still possible that WASP activates PLCg by acting as a scaffold? Again, please make the experiments internally consistent (e.g., use WASP-/- cells throughout as recipient cells) and explain the choice of Y291F as a control*.

We thank the reviewers for helping us clarify the overexpression results further.

It is clear that these experiments rely on dominant-negative action of overexpressed mutant versions of WASP. The overexpression of WASP constructs in human model T cell lines has previously been utilized as a successful strategy to assess WASP function (71). In human primary T cells, we find that the overexpressed constructs localize to the foci sites in a manner similar to WT WASP (Figure 4, Figure 11). Overexpression of WASP C resulted in a strong perturbation of patches as well as PLC γ1 phosphorylation, indicating that it is able to compete with endogenous protein for effector function. We didn’t observe a dominant negative effect with WASP Y291F, which lacks a potential Src family kinase SH2 binding site (Padrick, 2010) (Figure 4). The failure of this mutant to have an effect suggests that binding of Lck or Fyn, the major Src family kinases in T cells to WASP, is not required for the formation and function of foci. Furthermore, WASP Y291 phosphorylation has been shown to be dependent upon the binding of WASP to Cdc42 (Torres et al., 2003) while foci induction (Figure 11) as well as WASP recruitment at the T cell synapse can both occur independent of Cdc42 (14).

Author response image 4.(A) Distribution of overexpressed GFP-WASPY291F in synaptic plane. A representative cell from the results presented in Figure 4 is shown here. Human CD4 T cells were transfected with GFP-WASP291F and utilized to assess the effect of their overexpression on foci formation, as described in Figure 4. Note that GFP fluorescence (top left panel) is seen in a punctate form that colocalizes with actin Foci/phospho-PLCγ1 distribution. When assessed for overall intensity of GFP puncta at the synapse, Y291F shows comparable extent of recruitment as GFP-WT WASP (GFP-WT, 1.00±0.06, n=27; GFP-WASPY291F, 0.898±0.075, *p=0.332*). (B) Loss of Cdc42 does not reduce F-actin foci formation. Freshly isolated primary CD4 T cells from wild type (WT, left panels) or Cdc42 -/- mice (right panels) were incubated with 2C11 and ICAM1 containing bilayer for 2 min, fixed and stained with anti-phospho HS1 antibody and Alexa488-phalloidin. Cells were imaged using TIRFM, and images were quantified for foci formation (Graph on right). Note that lack of Cdc42 does not impair foci formation. This experiment was repeated twice with similar results. In the graph, n1=40, n2=24, p=0.632*.***DOI:**
http://dx.doi.org/10.7554/eLife.04953.038

Therefore, the reviewers are correct that it addresses a separate issue and it not a better control for the ∆C mutant than WT WASP and we thank the reviewers for calling this to our attention. We provide this rationale in the Results section and briefly discussed the implications of the results with this in mind. Complementary experiments based on testing the ability of different WASP mutants to rescue defects in the WASP-/- T cells would address the specificity of the phenotype in the WASP-/- T cells (6) and what parts of WASP are sufficient for function, perhaps by making chimeras with NWASP to map the unique regions of WASP that are required. We believe that these are excellent experiments, but strongly feel that they should be part of a future study to investigate unique roles of WASP and NWASP in T cells and are beyond the scope of this study.

*3) In*
Figure 3*, the authors determined whether TCR signaling is required for formation of the actin foci. They have compared “ICAM alone” and “ICAM + MHCp” conditions. It would be useful to include the “MHCp alone” condition as well to determine if integrin signaling is also involved*.

Sensitive antigen recognition requires adhesion molecules. In the absence of LFA- 1 or other adhesion molecules the amount of peptide -MHC required to generate TCR clusters is 10-100 fold higher. Consistently, when we utilized peptide-MHC alone bilayer at 100 molecules/µm2, we could not obtain synapses stable enough to withstand washing conditions required for phalloidin staining. In the literature, real time visualization of T cell engagement of beads coated with anti-CD3 only results in localized actin rich protrusions at the site of the contact (Husson et al., 2011). To dissect the selective participation of TCR and integrin (LFA-1) in foci induction and dynamics requires a different set of highly specialized tools that are beyond the scope of this study (please see the first paragraphs of the subheading “Unique requirement of WASP for foci generation” in the Discussion section). In a parallel studies, we have collaborated with the groups of Michael Sheetz and Lance Kam using micro-patterned TCR and integrin ligand dots to investigate the distinct biochemical contributions of LFA- 1 and TCR to the immunological synapse (Tabdanov et. al., submitted for a special issue of “Integrative Biology”). Using this system, we observed that the actin foci are indeed generated by TCR signaling, and LFA1–dependent signaling aids dissipation of f-actin from the foci to allow integrin dependent spreading. These data suggest that integrin shape foci dynamics, but are not required for their formation. We hope that in a final form of both papers that these complementary studies can be cross-referenced.

*4) The Arp2/3 inhibitor CK666 abolished 67% actin foci and 42% of lamellipodia and lamellar actin. While the foci are marginally more reduced, it is difficult to attribute the defects in calcium flux and NFAT translocation (*Figure 6*) to foci reduction rather than lamellipodia and lamellar actin reduction. It might be better to test these downstream signaling defects in WASP deficient cells because depleting WASP appears to be more selective for reducing foci and does not affect lamellar actin structures as much. Further, at least some of the results obtained using CK666 should be confirmed with a more selective approach based on Arp2/3 silencing*.

In WASP-/- T cells as well as the T cells from WAS patients, signaling events downstream of PLCγ1 activation have already been characterized extensively. These include defects in TCR-induced calcium ion flux, NFAT translocation and IL-2 transcription (please see the third paragraph of the Introduction and [11]; Cannon et al., 2004; [19]; [92]). These TCR-distal signaling defects were comparable to those reported for WASP-/- cells, when we treated cells with CK666 (Figure 6).

Akin to WASP deficiency, early TCR signaling is intact in CK666 treatment, distinguishing it from any general F-actin inhibitor, such as Latrunculin A or cytochalasin B, which profoundly inhibit early TCR signaling. Thus, even though CK666 is a broad-range inhibitor, its functional consequences highlight a stage -specific role of Arp2/3 and WASP in TCR signaling. It is indeed interesting that a loss of an additional 50% of total F-actin in CK666 treatment, as compared to WASP-/- cells, causes similar defects in late TCR signaling. The functional significance of the large lost fraction of total F-actin is unclear at the moment, however, it is unlikely to impact WASP-dependent calcium ion signaling cascade (Discussion section, paragraph 5).

The reviewers have brought up an excellent point regarding the specificity of CK666. Recently, a careful comparison of Arp3 shRNA and CK666 was carried out, where CK666 was found to phenocopy Arp3 knockdown (Bovellan et al., 2014). Arp3 shRNA or CK666 both reduced cortical actin by about 50%, very similar to our findings in T cells and thus validating our use of the inhibitor over genetic manipulation of Arp2/3 complex. Additionally, given the benefit that CK666 provides better temporal control of Arp2/3 complex inhibition compared to Arp2/3 knockdown, during live imaging of cells, we chose to utilize it for our experiments.

*5) The results showing the role of WASP on foci formation are obtained using fixed and permeabilized cells at a defined time point (2 minutes after beginning of culture). This makes it difficult to estimate the time of formation of these structures. It would be important to investigate the time kinetics of foci formation and of their migration towards cSMAC using time-lapse video microscopy to define whether the kinetics are altered (delayed) upon WASP/Arp2/3 silencing. It would also be interesting to monitor the kinetics of foci formation and of their lifetime in different areas of the synapse (independently of WASP silencing) in parallel with [Ca*^*2+*^*]i increases. This approach would allow to establish whether foci formation precedes or not the [Ca*^*2+*^*]i increase*.

This would be a valuable dataset to obtain, but we did not undertake this experiment and would like to explain our reasons.

We and others have previously shown that in WASP deficiency, the cell proliferation and cytokine secretion defects are of reduced amplitude, rather than delayed kinetics (Cannon et al. , 2004; [72]; [92]), since they manifest after prolonged culture of cells (30 min-24 hours). Our current data shows that the differences in F-actin foci between WT and WASP- /- or Arp2/3 inhibited cells are also of magnitude rather than kinetics. WASP-/- T cells fail to recover foci even after 1 hour of synapse formation (Figure 12). Similarly in CK666 treatment, foci are permanently lost from the synapse even after prolonged incubation with the bilayer (Figure 6—figure supplement 1), and do not recover even after 1 hour of incubation (data not shown). Thus, delayed kinetics of foci formation cannot explain the loss of foci in WASP-/- or CK666-treated cells.

Author response image 5.Prolonged Lack of foci in WASP-/- T cells. CD4 T cells from WT mice were isolated and labeled with CFSE (carboxyfluorescein diacetate, succinimidyl ester, 5μM, pseudocolored green). CFSE labeled WT cells were mixed with WASP-/- T cells, incubated with surface coated with 2C11 and ICAM1 for 10 min (upper panels) or 1 hour (bottom panels), fixed and stained with Alexa568-phalloidin (pseudocolored red). The synaptic contacts cells were then imaged using spinning disc confocal microscope. Note that while WT cells (CFSE positive, green) maintain foci at 1 hour post activation, WASP-/- T cells (arrowheads) persistently lack foci. This lack of foci is clearly visible in more than 85% of WASP-/- T cells, both at 10 min and 1 hour (n>48).**DOI:**
http://dx.doi.org/10.7554/eLife.04953.039

A real time demonstration of foci formation and calcium ion flux in live cell would be an ideal experiment, but it is extremely challenging to achieve it with the current imaging techniques. Using live cell video microscopy, we found that the TCR microclusters and F-actin foci form rapidly (within 4 second frame rate they appear together for first time in many instances: Figure 2—figure supplement 5, Video 6). From the results of Huse et al, we know that Calcium flux takes place ∼6 seconds after TCR ligands become available. The kinetic window between TCR microcluster formation, foci induction and calcium release is too small to be captured distinctively. Tracking individual foci and calcium flux is further complicated by the fact that synapse maturation is a rapid and highly dynamic process, where lamellar fluctuations occur continuously (Sims et al., 2007) and fresh microclusters/foci continue to form and migrate centripetally in the synaptic interface (Figure 2—figure supplement 5, Figure 6—figure supplement 3). Therefore, we would have to assay calcium ion dynamics at the sites of individual mobile foci. Additionally, calcium ions rapidly diffuse in the cytoplasm, preventing the recording of calcium ion fluxes at the site of single foci. Thus, given the challenging nature of the process for live cell imaging purpose, and the inadequacy of current imaging methodologies to achieve it, we were unable to pursue this question further.Author response video 1.Actin foci associate with TCR MC as soon as synapse forms. Human CD4 T cells were transfected with LifeAct-GFP (green) and incubated with SLB reconstituted with Alexa568-OKT3 (anti-CD3, red), and ICAM1. Cells were imaged live using TIRFM at the rate of 1 time frame per 5 sec. The movie shows sequence of events during the first 200 sec of synapse formation. Left panel shows F-actin dynamics using the raw images, while the panel on the right shows a time lapse from spatial frequency filtered F-actin images (Methods, Figure 1—figure supplement 1). The play rate of the movie is 33 times faster than the original image acquisition frame rate.**DOI:**
http://dx.doi.org/10.7554/eLife.04953.04010.7554/eLife.04953.040

*6) The authors propose that WASP pathway regulates actin foci formation, which in turn is instrumental for synaptic activation of PLCγ1. However, they do not show a mechanistic link between foci formation and PLCγ1. An alternative explanation of the data might be that the WASP pathway might control PLCγ1 activation. In turn PLCγ1 signaling might promote actin dynamics as suggested by a recent report (A. Brodovitch et al. J Immunol 2013; 191:2064-2071). To address this point, the authors should investigate the impact of PLCγ1 silencing (or of its pharmacological inhibition) on F-actin foci formation and the time kinetics of foci formation and of [Ca*^*2+*^*]i (see above)*.

The reviewers make an excellent point. The field has known for a long time that PLC isoforms are important for integrin function and, therefore, will affect synaptic maturation and function (Brodovitch et al., 2013), which in turn will facilitate PLCγ activation. While this feed-forward involvement of PLCγ is essential for synapse formation, it is WASP independent. Our current data show that eliminating WASP or inhibiting Arp2/3 complex both decrease, but do not eliminate PLCγ from TCR microclusters. Therefore, normal synapse formation in WASP deficient T cells is fully consistent with WASP independent PLCγ activation that is critical for synapse formation. We have now included it in the description of our model.

We did follow up on the reviewers’ suggestion to investigate the role of PLCγ in WASP pathway. We attempted to inhibit PLC pharmacologically using two different doses of U-73122, and monitored the effect of this manipulation on foci formation (Figure 13). Under these treatment conditions, synapse attachment and total actin polymerization and foci formation, were inhibited simultaneously (Figure 13). This is consistent with the results demonstrated by one of the authors previously (Heissmeyer et al., 2004) that PLC inhibitors such as U-73122 destabilize immunological synapses, particularly the integrin rich components. This reaffirms that a pool of PLC γ or its isoforms is also required for normal synapse establishment, and is likely to be intact in WASP-/- T cells where synapse formation is not impaired.

Author response image 6.(A) Effect of PLC inhibition on synapse formation and foci induction. Mouse primary CD4 T cells were treated with indicated concentrations of PLC inhibitor, U-73122, or inactive control U-73343, or DMSO alone (Control) for 5 min at room temperature, then with 2C11 and ICAM1 coated surface for additional 5 min in the presence of above-mentioned inhibitors. Cells were then fixed, stained with Alexa-488 phalloidin and anti-phospho PLCγ1 antibody, and imaged using TIRF microscopy. The graph shows synaptic levels of phospho-PLCγ1, total F-actin or foci in different treatment backgrounds, normalized to the values obtained in the case of DMSO control (n in control= 98, in U-73343= 91, in 0.1μM U-73122=113, and in 1μM U-73122= 17). The foci could not be analyzed in cells treated with 1μM U-73122, since synapse spreading was severely diminished along with suppressed F-actin polymerization and reduced synaptic attachment (Figure 13).In all of the above figures, scale bar is 5μm.**DOI:**
http://dx.doi.org/10.7554/eLife.04953.041